# Spatially bivariate EEG-neurofeedback can manipulate interhemispheric inhibition

**Masaaki Hayashi[1], Kohei Okuyama[2], Nobuaki Mizuguchi[3], Ryotaro Hirose[1], Taisuke Okamoto[1], Michiyuki Kawakami[2], Junichi Ushiba[4]***

[1]Graduate School of Science and Technology, Keio University, Kanagawa, Japan; [2]Department of Rehabilitation Medicine, School of Medicine, Keio University, Tokyo, Japan; [3]Research Organization of Science and Technology, Ritsumeikan University, Shiga, Japan; [4]Faculty of Science and Technology, Keio University, Kanagawa, Japan

**Abstract** Human behavior requires inter-regional crosstalk to employ the sensorimotor processes in the brain. Although external neuromodulation techniques have been used to manipulate interhemispheric sensorimotor activity, a central controversy concerns whether this activity can be volitionally controlled. Experimental tools lack the power to up- or down-regulate the state of the targeted hemisphere over a large dynamic range and, therefore, cannot evaluate the possible volitional control of the activity. We addressed this difficulty by using the recently developed method of spatially bivariate electroencephalography (EEG)-neurofeedback to systematically enable the participants to modulate their bilateral sensorimotor activities. Here, we report that participants learn to up- and down-regulate the ipsilateral excitability to the imagined hand while maintaining constant contralateral excitability; this modulates the magnitude of interhemispheric inhibition (IHI) assessed by the paired-pulse transcranial magnetic stimulation (TMS) paradigm. Further physiological analyses revealed that the manipulation capability of IHI magnitude reflected interhemispheric connectivity in EEG and TMS, which was accompanied by intrinsic bilateral cortical oscillatory activities. Our results show an interesting approach for neuromodulation, which might identify new treatment opportunities, e.g., in patients suffering from a stroke.

*For correspondence:
ushiba@bio.keio.ac.jp

## Editor's evaluation

The authors developed a new method for self-regulating interhemispheric inhibition via a novel brain-computer interface. The effect on interhemispheric inhibition was observed over and above modulating cortical excitability of a single hemisphere.

## Introduction

Projection neurons wire the brain over long distances and provide a network between different brain regions. In particular, both hemispheres are structurally connected by transcallosal projections and exhibit functional cross talks (*Hofer and Frahm, 2006*; *Meyer et al., 1995*); such interhemispheric interaction is essential for higher order cognitive and sensorimotor brain functions. An early argument for interhemispheric interaction was that dynamic interplay via the callosum not only allows for simple coordination of processing between the hemispheres but also has profound effects on attentional functioning (*Banich, 1998*). In the motor domain, neurons in the monkey primary motor cortex are firing bilaterally and motor signals from the two hemispheres interact during unimanual motor tasks (*Ames and Churchland, 2019*); in the human motor cortex, measuring these neural activities

via electroencephalography (EEG) and paired-pulse transcranial magnetic stimulation (TMS) recently revealed that the two hemispheres act together and the related cortical oscillatory activity influences the inhibitory interhemispheric brain network (*Picazio et al., 2014*; *Stefanou et al., 2018*).

A critical issue is whether a specific interhemispheric activity can be volitionally controlled. Previous studies have focused on inhibitory interhemispheric sensorimotor network, evaluated by interhemispheric inhibition (IHI), from movement-related manner (*Duque et al., 2007*; *Duque et al., 2005*; *Liang et al., 2014*; *Morishita et al., 2012*; *Murase et al., 2004*; *Nelson et al., 2009*), and passive neuromodulation effects by non-invasive brain stimulation techniques such as transcranial direct current stimulation (tDCS) or repetitive TMS (rTMS) (*Boddington and Reynolds, 2017*; *Gilio et al., 2003*; *Williams et al., 2010*). Due to the experimental limitations related to these observational or open-loop paradigms, much less is known about the effects of changes in sensorimotor activity patterns in both hemispheres on IHI, and the possibility of manipulating IHI. Therefore, the closed-loop paradigms to manipulate bilateral sensorimotor activity patterns and IHI should be of great value for the understanding of human sensorimotor brain function (*Figure 1A*).

As a recent neural manipulative tool, numerous studies have used a brain-computer interface (BCI)-based neurofeedback whereby participants learn to volitionally desynchronize and synchronize oscillatory sensorimotor rhythms (SMR-ERD/ERS) in the contralateral sensorimotor cortex (SM1) through visual and/or somatosensory feedback (*Ang and Guan, 2017*; *Chaudhary et al., 2016*; *Ramos-Murguialday et al., 2013*; *Soekadar et al., 2015b*). BCI-based neurofeedback is supported by the fact that the intensity of SMR-ERD in EEG represents states of high versus low excitability of not only the SM1 (*Neuper et al., 2006*; *Neuper and Pfurtscheller, 2001*; *Pfurtscheller et al., 2006*), but also the corticospinal descending pathway, as measured by the motor-evoked potential (MEP) amplitude (*Takemi et al., 2015*; *Takemi et al., 2013*). Because both hemispheres are structurally and functionally connected, it is likely that the balance of bilateral SMR-ERDs and transcallosal excitability states are linked, as indicated by common variation of conditioning MEP amplitude and IHI (*Ferbert et al., 1992*; *Ghosh et al., 2013*; *Ni et al., 2009*).

Here, we sought to investigate the association of IHI and the balance of SMR-ERDs to understand the inhibitory sensorimotor functions of interhemispheric interaction that may critically depend on the oscillatory brain activity in both hemispheres. Furthermore, if changes in the oscillatory brain activity influenced IHI, we tested whether it can be manipulated using a dual-coil paired-pulse TMS protocol (*Daskalakis et al., 2002*; *Ferbert et al., 1992*). To assess the association of IHI with ongoing oscillatory brain activity, we used a recently developed spatially bivariate BCI-based neurofeedback technique that allows volitional modulation of SMR amplitude in both hemispheres (*Hayashi et al., 2021*; *Hayashi et al., 2020*) and triggers the TMS pulses in real time at predetermined bilateral SMR amplitudes. Our system, therefore, overcomes previously unresolved experimental limitations of the ordinary observational or open-loop experimental paradigm. That is, our paradigm enables participants to control the bivariate sensorimotor excitability, and determines whether it is possible to manipulate the IHI magnitude.

Using the novel closed-loop bivariate (bi-hemispheric brain state-dependent) EEG-triggered dual-coil paired-pulse TMS system (bi-EEG-triggered dual-TMS), we evaluated the effects of different states of the targeted bidirectional up- or down-regulated one-sided hemisphere on effective inhibitory interhemispheric network expressed by IHI: (1) resting-state (REST), (2) during right finger motor imagery (MI) without visual feedback (NoFB), (3) high (HIGH), (4) middle (MID), and (5) low (LOW) excitability states of the ipsilateral SM1 (conditioning side) to the unilateral imagined hand movement while maintaining constant contralateral excitability during BCI-based neurofeedback (*Figure 1B*). Inspired by the findings that SMR-ERD was associated with corticospinal excitability amplitude (*Takemi et al., 2015*; *Takemi et al., 2013*), we hypothesized that the ipsilateral SMR-ERD to the imagined (right) hand is a potent up- or down-regulator of IHI from the ipsilateral (right) to the contralateral (left) hemisphere. This multimodal research provides strong evidence for the dynamic interplay between distinct regions underlying IHI through BCI-based neurofeedback, and may therefore form the basis for interhemispheric sensorimotor activity.

## Results

Values corresponding to each statistical figure are presented in each table (*Supplementary file 1*) for readability.

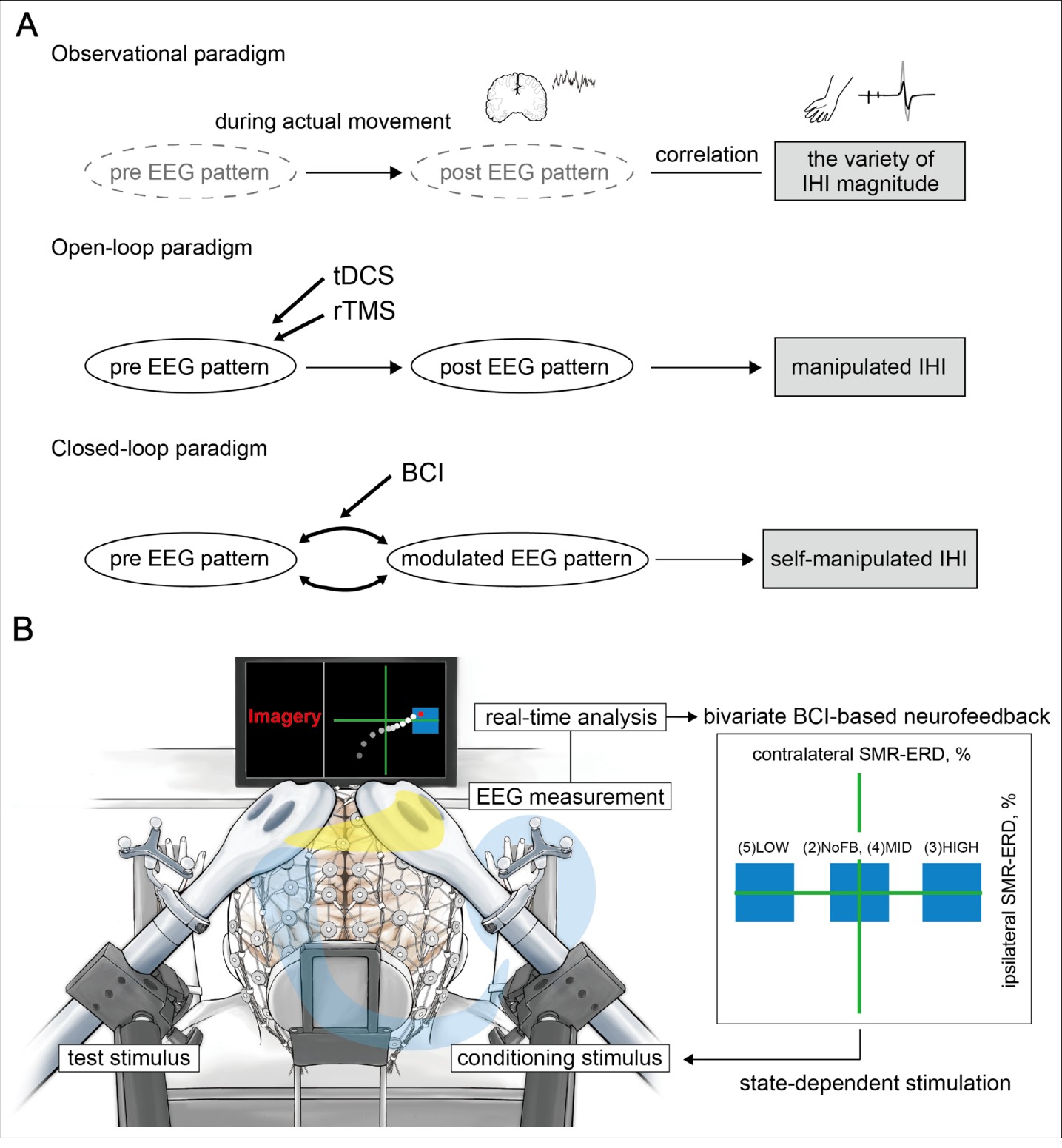

**Figure 1.** Conceptual illustration of the current study and experimental overview. (**A**) When a certain stimulus was input into the system, the brain was considered to vary with the state, resulting in interhemispheric inhibition (IHI) changes. The upper panel highlights the experimental limitations in the observational paradigm due to a variety of IHI magnitudes observed during actual movement. In this case, it is unclear whether the changes in electroencephalography (EEG) patterns in both the hemispheres would affect IHI. The middle panel indicates that it is unclear whether it is possible to manipulate inhibitory interhemispheric sensorimotor activity in the open-loop neuromodulation paradigm using transcranial direct current stimulation (tDCS) or repetitive transcranial magnetic stimulation (rTMS). The lower panel shows that specific EEG patterns are associated with IHI magnitude, and

*Figure 1 continued on next page*

*Figure 1 continued*

brain-computer interface (BCI)-based neurofeedback modulates the EEG activities. Therefore, if bilateral EEG patterns that underlie IHI are identified, we should be able to volitionally regulate the IHI magnitude via BCI-based neurofeedback, suggesting the possibility of plastic interhemispheric balancing. (**B**) The current bi-EEG-triggered dual-TMS experimental system involved spatially bivariate BCI-based neurofeedback that allows volitional modulation of EEG patterns only in a targeted hemisphere to enable us to verify our hypothesis. Different states of the targeted bidirectional up- and down-regulated ipsilateral hemisphere to the imagined hand while maintaining constant contralateral excitability were tested in the following states: (1) resting-state, (2) during motor imagery without visual feedback, (3) high, (4) middle, and (5) low excitability states. A blue target box, based on the predetermined SMR-ERDs, was displayed corresponding to each session. A cue signal was generated to trigger the conditioning stimulus when the signal reached the target box. The yellow line on the head represents the signal flow from the conditioning hemisphere that modifies the contralateral side through the corpus callosum, and the blue line represents the test stimulus toward the right hand.

## Data compliance

Participants completed all five sessions as follows: (1) resting-state (REST), (2) right finger MI without visual feedback (NoFB), (3) high excitability states of the ipsilateral SM1 during BCI-based neurofeedback (HIGH), (4) middle excitability states of the ipsilateral SM1 during BCI-based neurofeedback (MID), and (5) low excitability states of the ipsilateral SM1 during BCI-based neurofeedback (LOW). In the three neurofeedback sessions (i.e. HIGH, MID, and LOW), participants received visual feedback during right finger MI and tried to modulate the sensorimotor activity patterns. The NoFB session served as a baseline to examine the IHI magnitude without visual stimuli and volitional control. The REST session also served as a control to determine the intrinsic IHI magnitude at rest.

In all MIs-based sessions (NoFB, HIGH, MID, and LOW), participants subjectively confirmed that they were able to perform kinesthetic MIs of unilateral right hand index finger abduction. The EEG signals with amplitudes $\leq 100$ µV were consistently recorded throughout the sessions. Contralateral and ipsilateral SMR amplitudes to the imagined hand were greater during the resting epoch than during the MI epoch, indicating desynchronization of the sensorimotor neural activity induced by MI. The averaged power spectrum density from the C3 channel during the resting epoch showed peak frequencies at 8–13 Hz and 21–24 Hz across the participants (*Crone et al., 1998*; *Neuper et al., 2006*; *Pfurtscheller, 2001*). The averaged powers in the alpha and beta bands were $1.67 \times 10^{-4} \pm 0.34 \times 10^{-4}$ and $2.85 \times 10^{-5} \pm 0.54 \times 10^{-5}$ V$^2$ (mean ± SD), respectively. All procedures were well tolerated and no adverse events were noted.

To quantify the difficulties in each neurofeedback session, two indices were examined: (1) the number of triggered trials represents the BCI performance (i.e. success trials) since TMS was only triggered when the EEG signal exceeded the predetermined SMR-ERD threshold; (2) mean waiting time for a triggered event from task onset, which yields a range of 0–5 s (triggered TMS timing). The mean values of all triggered trials (±1 SD) of the neurofeedback sessions were 14.2 ± 4.7 trials and post-hoc paired t-tests following a one-way repeated measures ANOVA (rmANOVA) revealed no significant differences between the three sessions (all p>0.05; HIGH: 12.0 ± 5.0 trials, MID: 16.1 ± 3.8 trials, and LOW: 15.9 ± 5.4 trials). The mean waiting time for a triggered event from task onset (±1 SD) was 1.96 ± 0.81 s, and post-hoc paired t-tests following an rmANOVA showed no significant difference between the three sessions (all p>0.05; HIGH: 2.16 ± 0.76 s, MID: 2.05 ± 0.73 s, and LOW: 1.71 ± 0.94 s).

## Modulation effects of bilateral SM1 at EEG level

We tested whether participants could learn volitional up- or down-regulation of the ipsilateral sensorimotor excitability to the imagined right hand while maintaining constant contralateral sensorimotor excitability at the EEG level using a spatially bivariate BCI-based neurofeedback. For the contralateral SMR-ERD, a one-way rmANOVA for the sessions (five levels: REST, NoFB, HIGH, MID, and LOW) revealed a significant difference ($F_{(4,102)} = 4.81$, p=0.014, and $\eta^2$=0.16). Post-hoc two-tailed paired t-tests demonstrated no significant differences in the contralateral SMR-ERD between the neurofeedback sessions (HIGH-MID: Cohen's *d*=0.13, p=1.00; HIGH-LOW: Cohen's *d*=0.22, p=1.00; MID-LOW: Cohen's *d*=0.11, p=1.00), whereas significant differences were found between REST and other sessions (all p<0.05; *Figure 2A*). In contrast, after showing significant differences in the ipsilateral SMR-ERD between sessions ($F_{(4,102)} = 156.0$, p<0.001, and $\eta^2$=0.86), post-hoc two-tailed paired t-tests showed that ipsilateral SMR-ERD increased during the HIGH session (HIGH-MID: Cohen's *d*=3.12, p<0.001; HIGH-LOW: Cohen's *d*=6.13, p<0.001) and decreased during the LOW session (MID-LOW: Cohen's

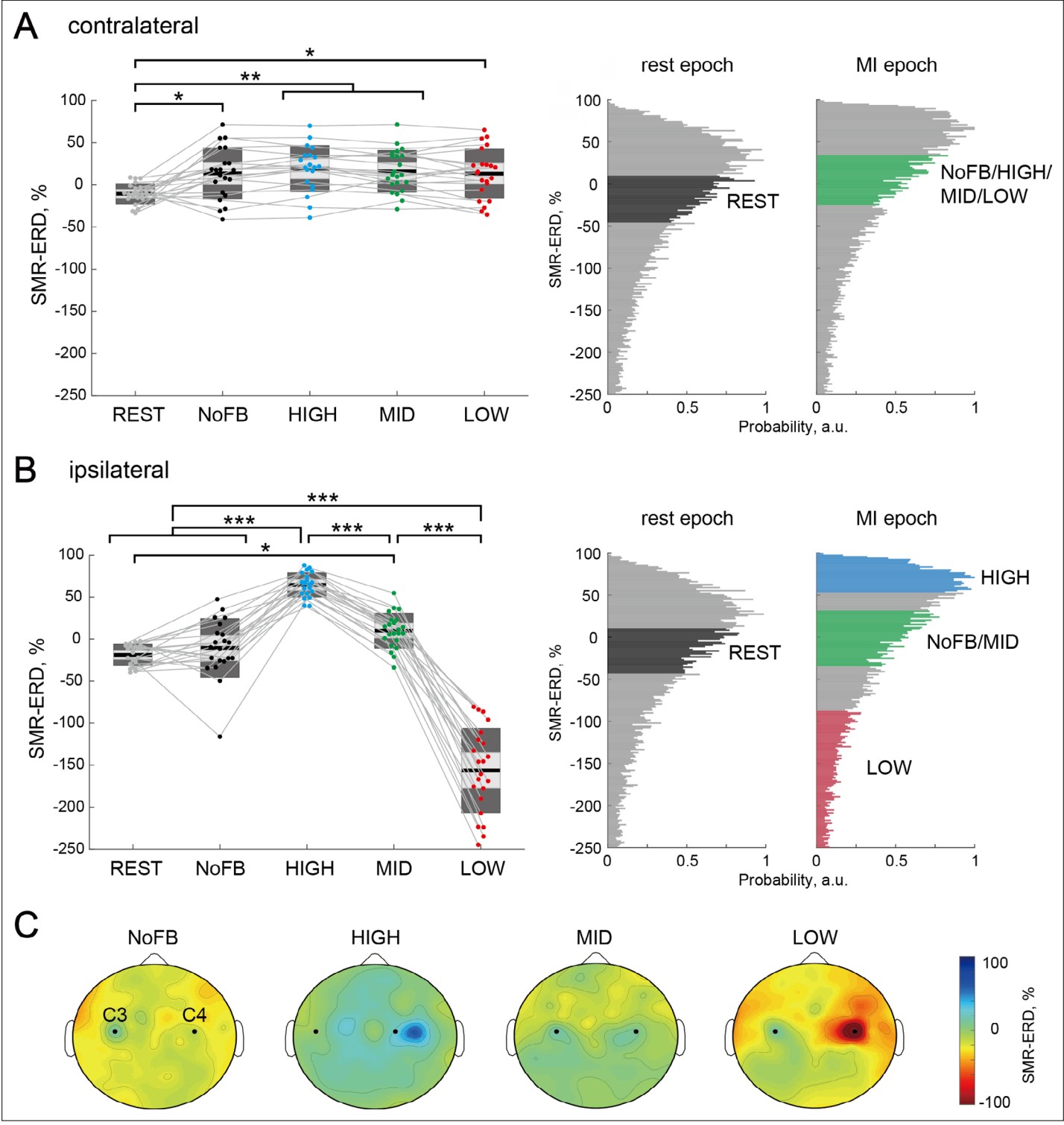

**Figure 2.** Target-hemisphere-specific modulation at the electroencephalography (EEG) level induced by spatially bivariate brain-computer interface (BCI)-based neurofeedback. (**A** and **B**) Modulation effect of the contralateral (left) and ipsilateral (right) SMR-ERDs to the imagined right hand, respectively. Individual participants are represented by colored plots and thin gray lines. The light gray box represents 1.96 SEM (95% CI) and dark gray box represents 1 SD. The black line indicates the group mean of the studied sample and colored plots indicate a single session. Positive values indicate desynchronization as compared to rest. Complete desynchronization to zero power in the frequency of interest translates to a 100% increase in SMR-ERD, whereas synchronization in the same band could theoretically be unlimited and allows for decreases in SMR-ERD>100%. The two right-sided panels represent the SMR-ERD distributions during rest and motor imagery (MI) epoch in the calibration session. Based on the SMR-ERD distributions in

*Figure 2 continued on next page*

*Figure 2 continued*

the contralateral and ipsilateral hemispheres, the target ranges of SMR-ERD during bi-EEG-triggered dual-transcranial magnetic stimulation (dual-TMS) system (each color) were set for each participant. (**C**) Spatial patterns of SMR-ERD during the MI epoch in each session (group mean). Large positive values (blue color) represent larger SMR-ERD. The black dots represent the C3 and C4 channels.

The online version of this article includes the following figure supplement(s) for figure 2:

**Figure supplement 1.** Target-hemisphere-specific modulation at the electroencephalography (EEG) level in frequency bands outside of a target alpha band.

*d*=4.44, p<0.001), revealing a significant bidirectional modulation compared to baseline sensorimotor endogenous activity (*Figure 2B*). The spatial patterns of SMR-ERD in each session are depicted in *Figure 2C*. We found that the SMR-ERDs were predominantly localized in bilateral parieto-temporal regions (C3 and C4 channels and their periphery), and strong ipsilateral SMR-ERD was observed in the HIGH session with constant contralateral SMR-ERD. Modulation effects of the contralateral and ipsilateral excitabilities (power attenuation) in the theta (4–7 Hz), low beta (14–20 Hz), high beta (21–30 Hz), and gamma (31–50 Hz) bands were shown in *Figure 2—figure supplement 1* (see also *Supplementary file 1*–Table 2).

## IHI curves at rest

To validate IHI measurement under bi-EEG-triggered dual-TMS setup, IHI curves were obtained at rest (*Figure 3A*) in 20 out of 24 participants, with conditioning stimulus (CS) of varying intensity (five different intensities, 100–140% of resting motor threshold [RMT], in steps of 10% RMT). A one-way rmANOVA for intensities (six levels: 0 [single test stimulus: TS-only], 100, 110, 120, 130, and 140% RMT) revealed significant difference in intensity ($F_{(5,109)} = 8.31$, p<0.001, $\eta^2$=0.28). Post-hoc two-tailed

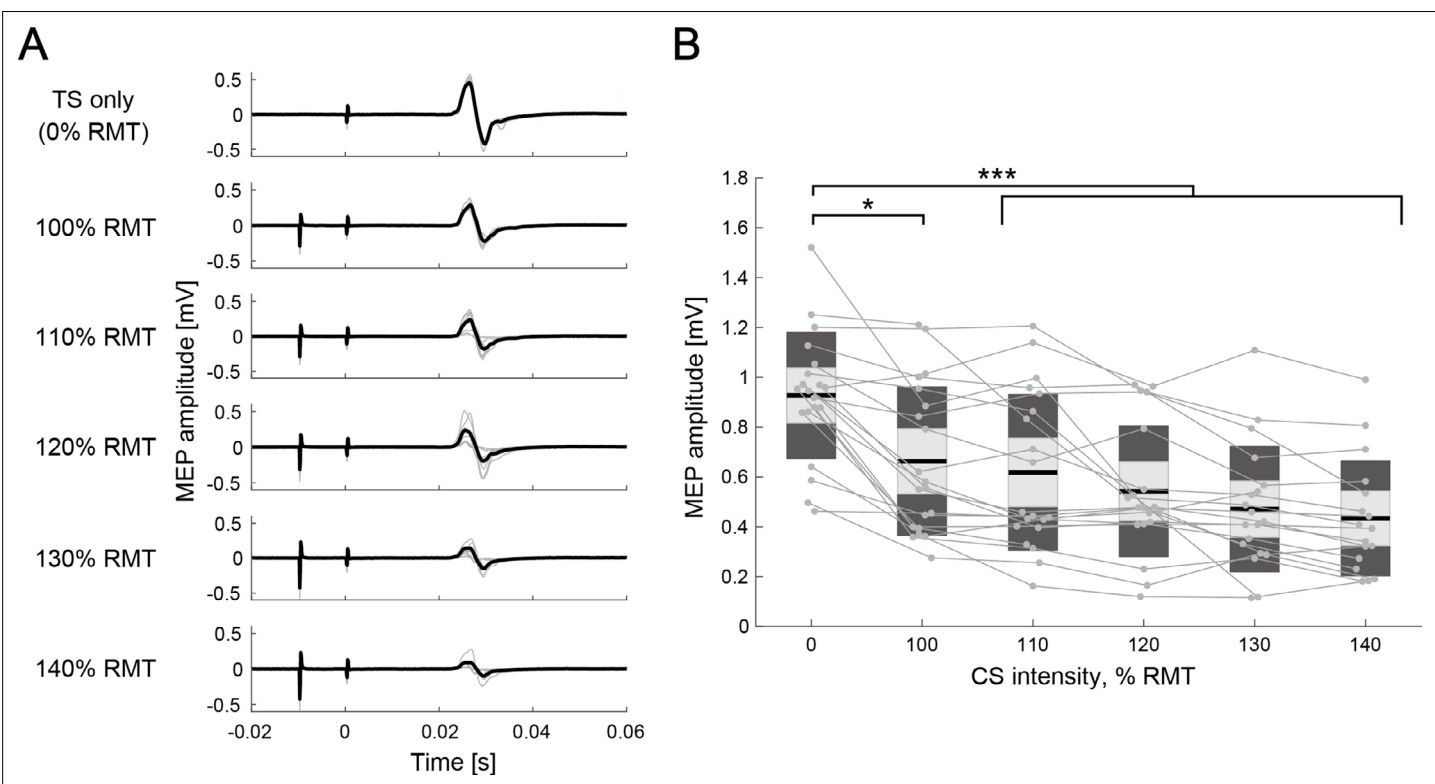

**Figure 3.** Interhemispheric inhibition (IHI) curves at rest. (**A**) Motor-evoked potential (MEP) amplitudes in the different intensity of conditioning stimulus (CS) of a representative participant. The thin gray lines represent each trial and black lines indicate the trial mean. (**B**) The IHI curves of the individual participants at rest are represented by thin gray plots and lines. The figure presents the individual data as an alternative to a box plot. The light gray box represents 1.96 SEM (95% CI) and dark gray box indicates 1 SD. The black line indicates the group mean. The y-axis indicates raw MEP amplitude against CS intensity (x-axis, in % resting motor threshold [RMT]). Dendrograms above the bars represent the results of the post-hoc analyses. *p<0.05 and ***p<0.001; all comparisons were Bonferroni corrected.

paired t-tests showed significant difference between TS-only and 100–140% of RMT (TS-only versus 100% RMT: p=0.039, TS-only versus 110–140% RMT: all p<0.001). There were no significant differences across CS intensities, while the size of MEP amplitude tended to be smaller for larger CS intensities. The IHI was approximately half of the maximum when the CS intensity was approximately 130% of RMT, which was compatible with a previous EEG-TMS experiment (*Stefanou et al., 2018*; *Tsutsumi et al., 2012*). In addition, the stimulus intensities of the left and right M1 to evoke MEP of 1 mV peak-to-peak amplitude from the relaxed right and left first dorsal interosseous (FDI) muscles ($SI_{1mV}$) were determined and used for the following dual-coil paired-pulse TMS setup. The average $SI_{1mV}$ of TS and CS were 124 ± 11% RMT and 132 ± 13% RMT, respectively. We also confirmed that the $SI_{1mV}$ of CS showed approximately 50% of the mean conditioned MEP over the mean unconditioned test MEP (*Figure 3B*).

## IHI manipulation via spatially bivariate BCI-based neurofeedback

Typical examples of MEP amplitude elicited by single TS (TS-only) and paired-pulse stimulation (CS + TS) of a representative participant are shown in *Figure 4A*. The manipulation range of IHI (i.e. the difference between HIGH and LOW sessions) was 32.6 ± 30.7% (Cohen's *d*=1.50) in all participants excluding 2 participants with data corruption (i.e. 22 participants). A one-way rmANOVA of the sessions (five levels: REST, NoFB, HIGH, MID, and LOW) revealed significant differences in IHI magnitude ($F_{(4,101)}$ = 6.85, p<0.001, $\eta^2$=0.22). Across the three BCI-based neurofeedback sessions (i.e. HIGH, MID, and LOW sessions) for the comparison of IHI magnitude, post-hoc two-tailed paired t-tests showed significant difference between HIGH and MID sessions (Cohen's *d*=0.94, p=0.025), and between HIGH and LOW sessions (Cohen's *d*=1.36, p<0.001), but not between MID and LOW sessions (Cohen's *d*=0.42, p=0.424; *Figure 4B*). In addition, we found disinhibition in NoFB session compared to REST session, although there was no statistical difference (Cohen's *d*=0.42, p=0.279); this trend is consistent with the previous studies, which reported that in healthy participants, IHI targeting the moving index finger leads to disinhibition before the movement onset to produce voluntary movement (*Duque et al., 2005*; *Murase et al., 2004*) and during isometric contraction (*Nelson et al., 2009*). Importantly, rmANOVA for the MEP amplitudes induced by TS-only revealed no significant differences between all sessions ($F_{(4,101)}$ = 2.44, p=0.104, $\eta^2$=0.09), suggesting that participants could learn to volitionally regulate the ipsilateral sensorimotor excitability while maintaining constant contralateral sensorimotor excitability (*Figure 4C*). A linear mixed-effect model that considered the contralateral SMR-ERD and ipsilateral SMR-ERD as fixed effect in the statistical regression model, and participants as random effect revealed a significant effect of the ipsilateral SMR-ERD (p<0.001) on IHI magnitude, but no main effect of the contralateral SMR-ERD or contralateral SMR-ERD × ipsilateral SMR-ERD interaction (p=0.828 and p=0.058, respectively).

To examine the IHI magnitude during neurofeedback, we further compared the IHI magnitude across sessions by calculating the percentage changes in IHI magnitude to baseline (i.e. NoFB session), and investigating the difference between REST, HIGH, MID, and LOW sessions. A one-way rmANOVA for sessions (four levels: REST, HIGH, MID, and LOW) revealed significant differences ($F_{(3,81)}$ = 8.32, p<0.001, $\eta^2$=0.24). Post-hoc two-tailed paired t-tests showed significant differences between HIGH and MID sessions (Cohen's *d*=1.06, p=0.017), and between HIGH and LOW sessions (Cohen's *d*=1.42, p<0.001), but not between MID and LOW sessions (Cohen's *d*=0.45, p=0.734; *Figure 4— figure supplement 1*). The normalized results were compatible with the results without normalization presented in *Figure 4*.

To further examine the influence of spontaneous SMR fluctuations on IHI, non-triggered TMS trial (referred to as the failed trial) was delivered in random timing ranging from 5.5 to 6 s during the MI epoch. For example, data from HIGH session in *Figure 4—figure supplement 2* were collected from MID and LOW sessions, with the target ranges of SMR-ERD in HIGH session (*Figure 4—figure supplement 2*). A two-way rmANOVA for sessions (three levels: HIGH, MID, and LOW) and trials (two levels: triggered TMS trials and non-triggered TMS trials) revealed a significant main effect for sessions ($F_{(2, 117)}$ = 5.08, p=0.008, and $\eta^2$=0.08), but no main effect for trials ($F_{(1, 117)}$ = 0.32, p=0.575, and $\eta^2$<0.01) and for interaction ($F_{(2, 117)}$ = 1.82, p=0.167, and $\eta^2$=0.03). Thus, no significant difference in IHI between the volitional control of SMR-ERDs in the closed-loop environment differs from spontaneous SMR fluctuations was observed. Post-hoc two-tailed paired t-tests showed significant differences between the HIGH and MID sessions (Cohen's *d*=0.94, p=0.017), and between HIGH and LOW sessions (Cohen's

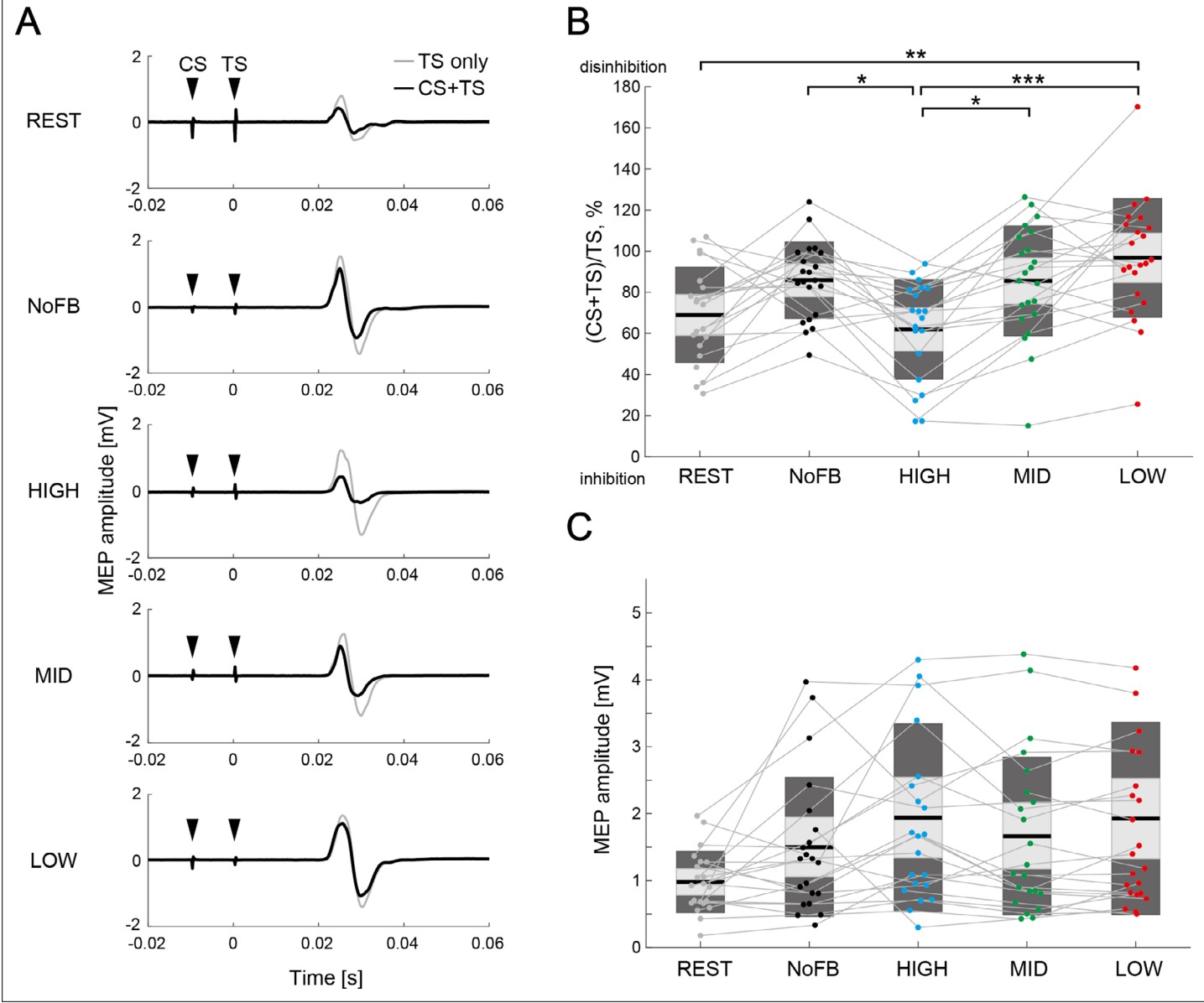

**Figure 4.** Comparison of interhemispheric inhibition (IHI) magnitude. (**A**) Typical examples of mean motor-evoked potential (MEP) amplitudes elicited by single TS (TS-only; light gray color) and paired-pulse stimulation (conditioning stimulus [CS] + TS; black color) in a representative participant. The black arrows represent the stimulus timings of CS and TS. (**B**) The IHI magnitudes of the individual participants are represented by colored plots and thin gray lines. The light gray box represents 1.96 SEM (95% CI) and dark gray box represents 1 SD. The black line indicates the group mean of the studied sample and colored plots represent a single session. Lower values represent greater inhibitory effect from the ipsilateral hemisphere to the imagined right hand. Dendrograms above the bars represent the results of the post-hoc analyses. *p<0.05, **p<0.01, and ***p<0.001; all comparisons were Bonferroni corrected. (**C**) The figure shows MEP amplitude elicited by a single TS (TS-only). No significant difference in MEP amplitude was observed between sessions (all p>0.05).

The online version of this article includes the following figure supplement(s) for figure 4:

**Figure supplement 1.** Comparison of the percentage change in interhemispheric inhibition (IHI) magnitude based on values in NoFB session.

**Figure supplement 2.** Comparison of interhemispheric inhibition (IHI) magnitudes in triggered and non-triggered transcranial magnetic stimulation (TMS) trials.

**Figure supplement 3.** Relationship between the increase in interhemispheric inhibition (IHI) and left motor-evoked potential (MEP) amplitude elicited by conditioning stimulus (CS).

**Figure supplement 4.** Comparison of interhemispheric inhibition (IHI) magnitude for control muscle.

*d*=1.36, p<0.001) in the triggered TMS trials. In contrast, in the non-triggered TMS trials, post-hoc two-tailed paired t-tests showed no significant differences between HIGH and MID sessions (Cohen's *d*=0.20, p=1.00), and between HIGH and LOW sessions (Cohen's *d*=0.28, p=1.00).

Although we revealed that modulating the excitability of the right hemisphere changes IHI magnitude from the right to the left hemisphere, the specificity of the IHI manipulation has not been proved yet. We presented additional results related to changes in corticospinal excitability as assessed by MEPs in the left FDI in response to the CS over the right hemisphere (*Figure 4—figure supplement 3B*). To investigate whether changes in IHI from the right to the left hemisphere and changes in corticospinal excitability in the right hemisphere are independent phenomena, a generalized linear model with the percentage changes of left MEP amplitude as a covariate of no interest was applied. This approach tests whether the effect of IHI manipulation survives even if the covariate is added. During the triggered TMS trials, a generalized linear model for sessions (three levels: HIGH, MID, and LOW) revealed a significant main effect for sessions ($F_{(3, 60)}$ = 12.75, p<0.001, and $\eta^2$=0.44). Post-hoc two-tailed paired t-tests in the increase in IHI showed significant differences between HIGH and MID sessions (Cohen's *d*=1.06, p<0.001), and between HIGH and LOW sessions (Cohen's *d*=1.42, p<0.001). During the non-triggered TMS trials, a generalized linear model for sessions (three levels: HIGH, MID, and LOW) revealed no main effect for sessions ($F_{(3, 60)}$ = 1.13, p=0.319, and $\eta^2$=0.10). Therefore, we successfully demonstrated that IHI changes greater than the variance explained by the CS effect and manipulation is not simply an epiphenomenon.

In the control muscle (abductor digiti minimi, ADM), such observed modulation was not driven. A one-way rmANOVA for sessions (five levels: REST, NoFB, HIGH, MID, and LOW) revealed significant differences ($F_{(4,101)}$ = 2.51, p<0.048, and $\eta^2$=0.12). Across the three BCI-based neurofeedback sessions (i.e. HIGH, MID, and LOW sessions) for the comparison of IHI magnitude, post-hoc two-tailed paired t-tests showed no significant difference between sessions (HIGH-MID: Cohen's *d*=0.10, p=1.00; HIGH-LOW: Cohen's *d*=0.05, p=1.00; MID-LOW: Cohen's *d*=0.15, p=1.00), whereas significant difference was observed between REST and NoFB sessions (Cohen's *d*=0.45, p=0.024; *Figure 4—figure supplement 4A*). Statistical analysis for TS-only revealed no significant differences in MEP amplitude between the sessions ($F_{(4,101)}$ = 0.62, p=0.649, and $\eta^2$=0.03; REST: 0.54 ± 0.51 mV, NoFB: 0.50 ± 0.39 mV, HIGH: 0.73 ± 0.61 mV, MID: 0.59 ± 0.50 mV, and LOW: 0.65 ± 0.51 mV; *Figure 4—figure supplement 4B*), indicating that IHI manipulation occurred only in the targeted muscle.

## Associations between IHI magnitude and bilateral EEG patterns

To examine the association between IHI magnitude and bilateral EEG patterns, we performed within- and across-participant correlation analyses, respectively. In the within-participant correlation analysis between IHI magnitude and bilateral SMR-ERDs in each participant, 7 of the 22 participants showed a significant correlation between IHI magnitude and ipsilateral SMR-ERD (r = –0.307 ± 0.252 [mean ± SD]), but not between IHI magnitude and contralateral SMR-ERD in three neurofeedback sessions (r = –0.059 ± 0.298 [mean ± SD]). A chi-squared test was performed to determine the overall significance of the 22 individual correlations between IHI magnitude and contralateral or ipsilateral SMR-ERDs. A comparison of the proportions of significant results obtained from the ipsilateral (7 out of 22 participants) and contralateral (1 out of 22 participants) SMR-ERDs with the proportion expected due to chance (i.e. alpha level = 0.05) showed statistically significant differences (chi squared value = 5.50, p=0.019 [Fisher's exact test: p=0.046]). The distributions of correlation coefficients across all participants were shown in *Figure 5A*.

In the across-participant correlation between IHI magnitude and contralateral or ipsilateral SMR-ERDs, we performed a repeated measures correlation analysis (*Bakdash and Marusich, 2017*). We found a significant correlation in the ipsilateral SMR-ERD ($r_{rm}$=–0.436, p=0.004; *Figure 5B*), but not in the contralateral SMR-ERD ($r_{rm}$=0.008, p=0.960; *Figure 5C*).

To analyze the associations between IHI magnitude and bilateral neural network from another perspective, interhemispheric functional connectivity during MI was examined. Interhemispheric functional connectivity was quantified as a Network-intensity measure (*Hayashi et al., 2020*), which is calculated from the corrected imaginary part of coherence (ciCOH) (*Ewald et al., 2012*; *Vukelić and Gharabaghi, 2015*). For the Network-intensity$_{MI}$, a one-way rmANOVA for the sessions (five levels: REST, NoFB, HIGH, MID, and LOW) revealed significant differences ($F_{(4,102)}$ = 3.04, p=0.020, and $\eta^2$=0.10). Post-hoc two-tailed paired t-tests in all five sessions demonstrated significant differences

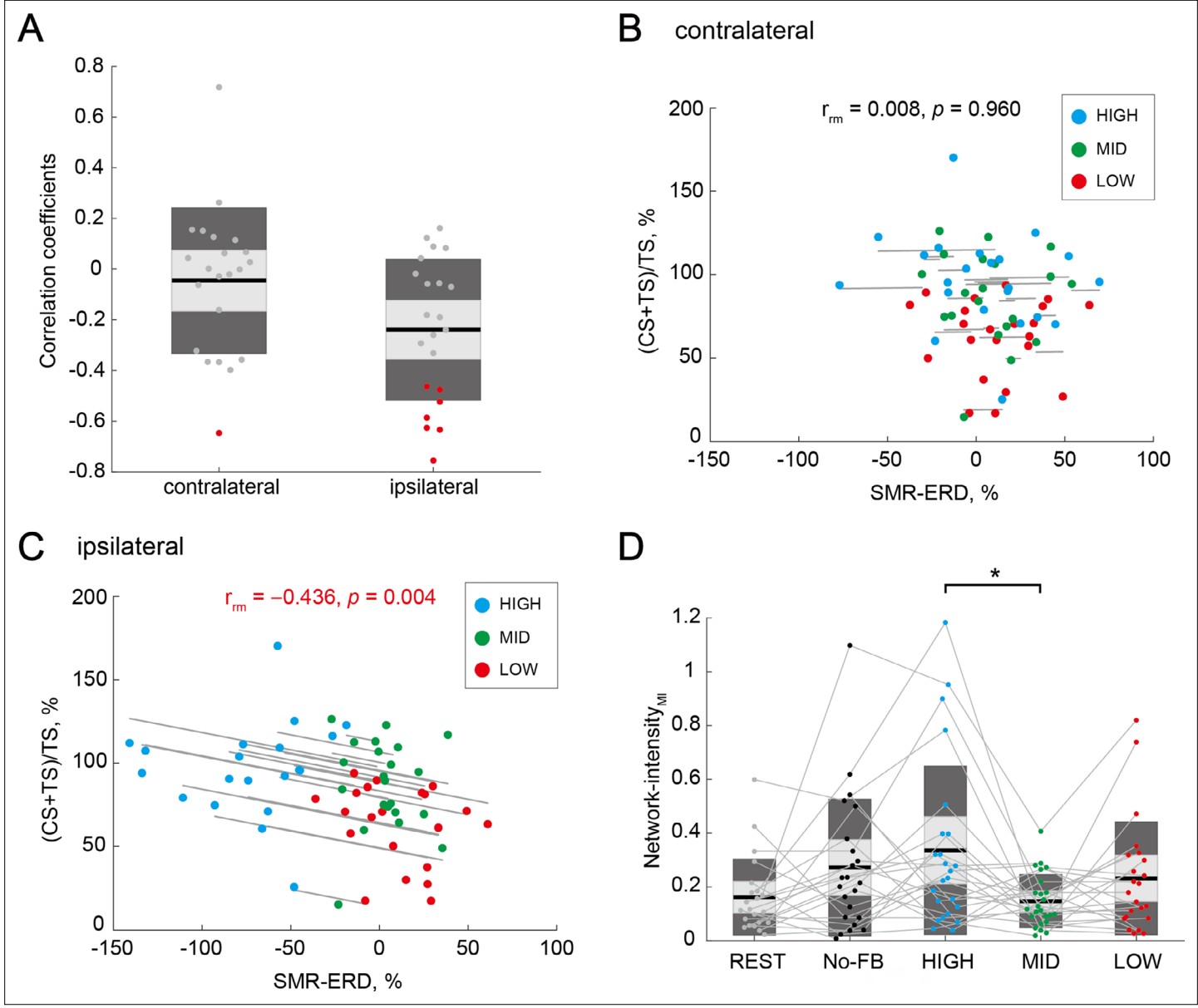

**Figure 5.** Associations of interhemispheric inhibition (IHI) magnitude and bilateral electroencephalography (EEG) patterns. (**A**) Distributions of within-participant correlation coefficients in all participants. Red dots represent individuals with a significant correlation between IHI magnitude and contralateral or ipsilateral SMR-ERDs, respectively. (**B** and **C**) Repeated measures correlations between IHI magnitude and contralateral or ipsilateral SMR-ERDs, respectively. Dots represent the mean value of a single session of each individual. Only the ipsilateral SMR-ERD and IHI magnitude were significantly correlated; high sensorimotor excitability state in the ipsilateral hemisphere would induce stronger inhibition from the ipsilateral hemisphere to the imagined hand. (**D**) Comparison of Network-intensity across sessions. Dendrograms above the bars represent the results of the post-hoc analyses. *p<0.05; all comparisons were Bonferroni corrected. There was a significant difference in interhemispheric functional connectivity between HIGH and MID sessions, but not between HIGH and LOW sessions.

The online version of this article includes the following figure supplement(s) for figure 5:

**Figure supplement 1.** Associations of interhemispheric inhibition (IHI) magnitude and bilateral electroencephalography (EEG) patterns in frequency bands outside of a target alpha band.

between HIGH and MID sessions (Cohen's $d$=0.83, p=0.033; *Figure 5D*). We found no significant difference between HIGH and LOW sessions (Cohen's $d$=0.40, p=1.00; *Figure 5D*), suggesting that interhemispheric functional connectivity did not reflect the excitatory or inhibitory activity. Associations of IHI magnitude and contralateral or ipsilateral EEG patterns in the theta (4–7 Hz), low beta

(14–20 Hz), high beta (21–30 Hz), and gamma (31–50 Hz) bands were shown in *Figure 5—figure supplement 1* (see also *Supplementary file 1*–Table 10).

## Individual characteristics associated with the manipulation capability of IHI

Finally, we investigated the neural characteristics associated with the manipulation capability of IHI calculated from the percentage change between HIGH and LOW sessions using correlation-based analysis (*Figure 6A*). In the relationships between the manipulation capability of IHI and IHI magnitude in REST session ($IHI_{rest}$), we found a significant correlation (r = –0.447, p=0.044), indicating that participants with greater IHI at rest were able to strongly manipulate the IHI (*Figure 6B*). For the relationships between resting-state effective inhibitory interhemispheric network assessed by $IHI_{rest}$ and interhemispheric functional connectivity at the EEG level (Network-intensity$_{rest}$), participants with greater $IHI_{rest}$ showed larger Network-intensity$_{rest}$ (r=0.547, p=0.013; *Figure 6C*). Furthermore, we verified whether $IHI_{rest}$ may be associated with intrinsic EEG profiles in NoFB sessions including bilateral SMR-ERDs. We found a significant correlation between $IHI_{rest}$ and ipsilateral SMR-ERD during MI (r = –0.619, p=0.004), but not between $IHI_{rest}$ and contralateral SMR-ERD during MI (r = –0.283, p=0.228).

In addition to the alpha band, the beta band is also a well-established EEG signature of motor execution and imagery (*Crone et al., 1998*; *Pfurtscheller, 2001*). To verify whether resting-state functional connectivity in other frequency bands (i.e. outside of feedback frequency) was associated with IHI, an across-participant Pearson's correlation was calculated. In the across-participant correlations between $IHI_{rest}$ and Network-intensity$_{rest}$ in the theta (4–7 Hz), low beta (14–20 Hz), high beta (21–30 Hz), and gamma (31–50 Hz) bands, we found significant correlation in the high beta (r=–0.618, p=0.004), but not in theta (r=–0303, p=0.194), low beta (r=–0.229, p=0.331), and gamma (r=–0.390, p=0.089) bands (*Figure 6—figure supplement 1A*). To attenuate the aberrant effect of values in some especially low Network-intensity$_{rest}$ or high (CS + TS)/TS at rest (outliers) in the theta, low-beta, high-beta, and gamma bands, the jackknife correlation and the bias were calculated (theta: r=–0.207, bias = 2.02; low-beta: r=–0.121, bias = 3.65; high-beta: r=–0.528, bias = –4.09; gamma: r=–0.310, bias = 0.06). In addition, the correlation between $IHI_{rest}$ and the contralateral or ipsilateral EEG patterns in frequency bands outside of a target alpha band were in *Figure 6—figure supplement 1B and C*, respectively.

## Discussion

In the present study, we aimed to determine whether it is possible to manipulate the IHI magnitude and uncover related neural activity behind IHI modulation, while controlling bilateral SMR-ERDs via spatially bivariate BCI-based neurofeedback. This was the first study to show IHI-state manipulation with a large dynamic range induced by volitional variation in ipsilateral sensorimotor excitability expressed by SMR-ERD. In addition, resting-state interhemispheric networks at the TMS ($IHI_{rest}$) and EEG (interhemispheric Network-intensity$_{rest}$) levels were associated with the manipulation capability of IHI magnitude.

Previous neurofeedback studies have demonstrated that it is possible to gain voluntary control over central nervous system activity without externally administered interventions, if appropriate neurofeedback is embedded in a reinforcement learning task, such as food rewards for animals (*Engelhard et al., 2013*; *Fetz, 2013*) and visually rewarding stimuli for humans (*Thompson et al., 2009*). However, conventional BCI-based neurofeedback of the SMR signal from one hemisphere does not guarantee spatial specific activation of the sensorimotor region in the targeted hemisphere (*Buch et al., 2008*; *Caria et al., 2011*; *Soekadar et al., 2015a*) because sensorimotor activities in the left and right hemispheres potentially influence one another, making IHI manipulation difficult. Using spatially bivariate BCI-based neurofeedback enables participants to volitionally increase or decrease (bidirectional) the ipsilateral sensorimotor excitability, while maintaining constant contralateral sensorimotor excitability. Ipsilateral SMR-ERD to the imagined hand (but not contralateral), reflecting IHI magnitude (*Figure 2B*) and their significant correlation (*Figure 5C*), were explained by previous studies (*Haegens et al., 2011*; *Khademi et al., 2018*; *Kraus et al., 2016*; *Madsen et al., 2019*; *Naros et al., 2020*; *Ros et al., 2010*; *Sauseng et al., 2009*; *Takemi et al., 2013*; *Thies et al., 2018*; *Zarkowski et al., 2006*). The in vivo cortical recordings in monkeys revealed that pericentral alpha power was inversely related with the

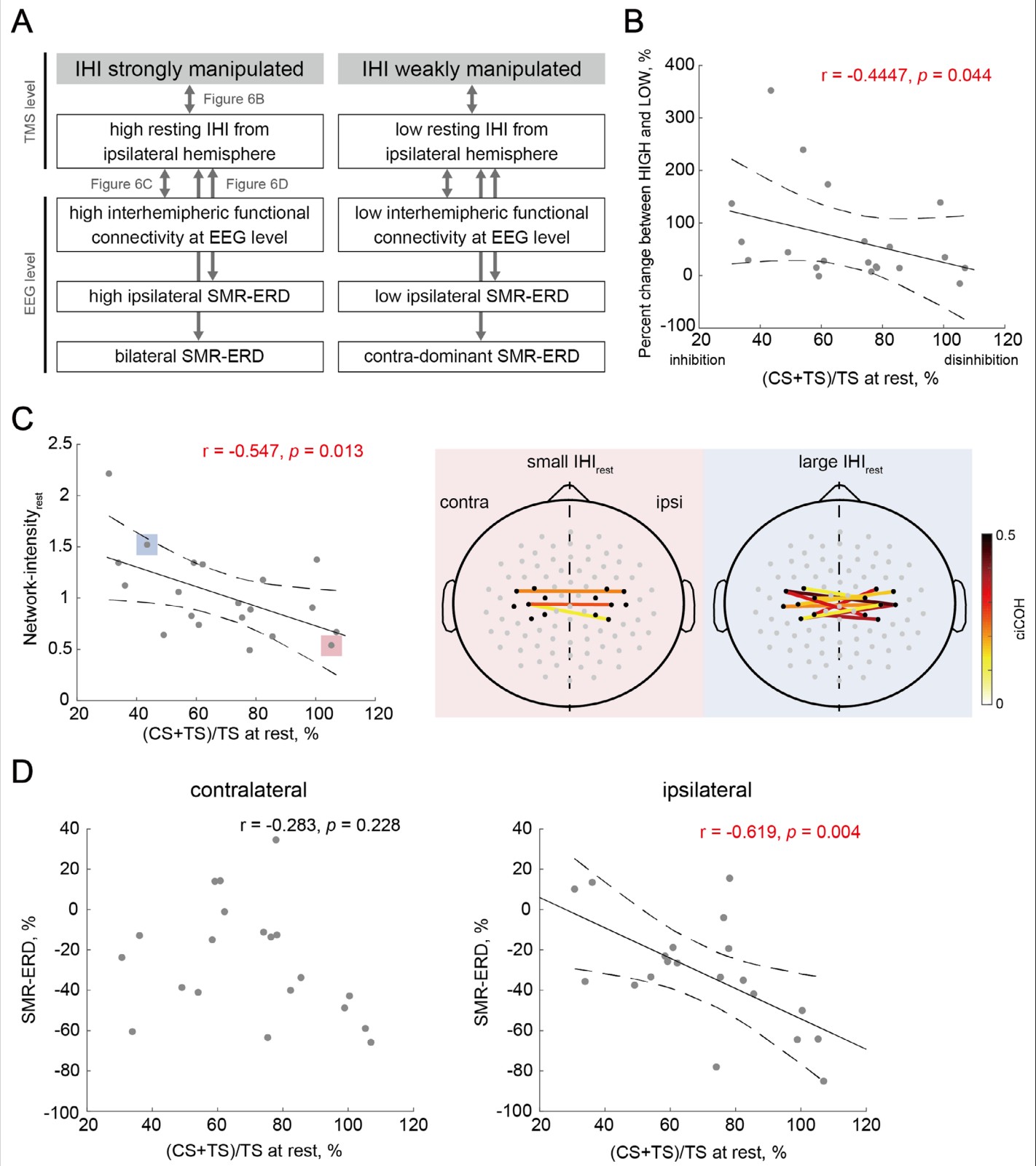

**Figure 6.** Individual characteristics associated with the manipulation capability of interhemispheric inhibition (IHI). (**A**) Overview of the relationships between biomarkers from electroencephalography (EEG) and transcranial magnetic stimulation (TMS) levels to probe the individual signatures for strong versus weak manipulation of IHI. Arrows correspond to a single panel. (**B**) Across-participant correlations between the manipulation capability of IHI and intrinsic IHI magnitude at rest. Dots represent a single participant. Solid and dotted lines represent the estimated linear regression and 95%

*Figure 6 continued on next page*

*Figure 6 continued*

CI, respectively. Participants with greater IHI at rest were able to strongly manipulate IHI. (**C**) The left-sided panel shows across-participant correlations between IHI at rest and resting-state Network-intensity between bilateral sensorimotor cortex (SM1). The two right-sided panels indicate the significant interhemispheric connections ('Connectivity analysis' in Materials and methods) of the two representative participants with small and large IHI$_{rest}$, respectively. The solid lines indicate a significant connection, and large positive values (dark red color) represent strong connections. The black dots around bilateral SM1 denote the seed channels, C3 or C4, and six neighboring channels. The gray dots represent other EEG channels. (**D**) Across-participant correlations between IHI at rest and EEG profiles showed significant correlation between IHI$_{rest}$ and ipsilateral SMR-ERD, but not between IHI$_{rest}$ and contralateral SMR-ERD.

The online version of this article includes the following figure supplement(s) for figure 6:

**Figure supplement 1.** Neural characteristics depending on the manipulation capability of interhemispheric inhibition (IHI) in frequency bands outside of a target alpha band.

normalized firing rate in the sensorimotor regions (*Haegens et al., 2011*). In humans, several studies using single-pulse TMS of the motor hand area combined with EEG tested how ongoing pericentral oscillatory activity impacts corticomotor excitability reflected by the MEP amplitude. These EEG-TMS studies found a negative relationship between pre-stimulus alpha power and MEP amplitude through both offline (*Sauseng et al., 2009*; *Takemi et al., 2013*; *Zarkowski et al., 2006*) and online (*Madsen et al., 2019*) approaches, whereas the opposite of that was also found (*Thies et al., 2018*). Furthermore, another EEG-TMS study using neurofeedback demonstrated that the endogenous suppression of alpha rhythms at rest can produce robust increase in MEP amplitude and decrease in short-interval intracortical inhibition (*Ros et al., 2010*). Similarly, in the beta band, some evidences were revealed that a negative relationship between beta power and MEP amplitudes (*Khademi et al., 2018*; *Kraus et al., 2016*; *Naros et al., 2020*). As for IHI through the transcallosal fiber, it is predominantly regulated through direct postsynaptic mechanisms in the apical dendritic shafts of pyramidal neurons and a specific cortical microcircuitry mediated by dendritic GABA$_B$ receptors in inhibitory interneurons (*Palmer et al., 2012*). Additionally, consistent with this phenomenon, down-regulation of ipsilateral SMR-ERD, e.g., may influence the excitation of the ipsilateral transcallosal pyramidal neuron followed by disinhibition of the contralateral inhibitory interneurons during unilateral upper limb MI. Therefore, pronounced changes in cortical mechanism despite the absence of sensory input and constant MEP amplitude in the imagined right hand elicited by single TS suggest that the modulation of IHI magnitude from the ipsilateral (right) to the contralateral (left) hemisphere was at least partly of cortical, rather than spinal origin. These neural mechanisms can be explored further by using triple pulse procedures (*Ni et al., 2011*) in which (dis)inhibition can be directly measured as the modulation of short-interval intracortical inhibition.

The potential contribution of resting-state IHI magnitude and interhemispheric functional connectivity in the alpha and high beta bands to the current results of manipulation capability in IHI (*Figure 6B and C* and *Figure 6—figure supplement 1A*) can be speculated by considering the previous studies. Inter-regional communication is accompanied by synchronized oscillations in different brain regions (*Fries, 2005*; *Varela et al., 2001*), and this synchronization can be evaluated by functional connectivity. Because both hemispheres are structurally connected by transcallosal projection and exhibit functional cross talks (*Hofer and Frahm, 2006*; *Meyer et al., 1995*), the manipulation capability of IHI is likely associated with a structural connectivity as well as local oscillatory power entrainment. As for frequency band outside of neurofeedback target, it is well known that the intensity of SMR-ERD in the beta band reflects the sensorimotor cortical excitability and corticomuscular activation (*Hussain et al., 2019*; *Schulz et al., 2014*). This multimodal EEG-TMS approach proves that the bilateral alpha and beta activities observed in the current study served to modulate the functional interhemispheric interaction over motor cortices.

For the other modalities, e.g., real-time fMRI-based neurofeedback to a single region of interest (*Sitaram et al., 2017*; *Weiskopf et al., 2004*) or inter-regional functional connectivity (*Liew et al., 2016*; *Pereira et al., 2019*) can be used for volitional modulation of neuronal connectivity and could serve as a possible therapeutic tool for motor or cognitive training in diseases related to impaired inter-regional connectivity. However, even fMRI-based neurofeedback with high spatial resolution, based on blood oxygenation level-dependent signal, cannot easily distinguish between the excitatory and the inhibitory activities from each region (*Moon et al., 2021*). Additionally, neurofeedback using interhemispheric functional connectivity has the possibility to counteract the modulation of

IHIs from the right to left hemisphere and from the left to right hemisphere. Our results showed no significant difference in Network-intensity during MI between HIGH and LOW sessions (*Figure 5D*), indicating that interhemispheric functional connectivity at the EEG level was not changed, whereas IHI was modulated. This may indicate that IHI and Network-intensity simply reflect the activity of different neural populations; interhemispheric projections act via surround/lateral inhibition in the sensory and motor cortices (*Carson, 2020*), while the synchrony at the EEG level reflects changes in the activity of local interactions between pyramidal neurons and interneurons in the thalamocortical loops (*Pfurtscheller and Lopes da Silva, 1999*). Furthermore, it is possible that Network-intensity reflects the degree of bidirectional synchrony between the two motor cortices, while IHI reflects one-way (right-to-left) inhibitory effects. Therefore, it is unclear whether fMRI-based neurofeedback using interhemispheric functional connectivity is linked to ipsilateral excitability, and whether it modulates IHI magnitude.

Interhemispheric activity is also passively modulated by externally administered interventions, e.g., tDCS or rTMS over the motor cortices (*Gilio et al., 2003*; *Pena-Gomez et al., 2012*; *Williams et al., 2010*). Although such tools have neuromodulation efficacy, their long-term sustained effects are often limited and they do not have spatial specificity due to the remote effects (*Di Pino et al., 2014*; *Notturno et al., 2014*; *Weiskopf et al., 2004*). Conversely, under the conscious selflearning environment, volitional control of MEP amplitudes is retained for at least 6 months without further training (*Ruddy et al., 2018*), which supports the future prediction of long-term efficacy of IHI manipulation. In addition, we successfully demonstrated that IHI changes greater than the variance explained by the CS effect. Based on our findings, further experiments may help to better understand the pathway specificity. To further confirm whether the participants learned pathway-specific IHI manipulation, the CS effect should be kept constant across the conditions by adjusting the CS intensity to evoke left MEP of 1 mV peak-to-peak amplitude. Furthermore, the effect size of IHI manipulation in the current study (Cohen's $d$=1.50) was comparable with that of representative tDCS (Cohen's $d$=1.55) (*Williams et al., 2010*) and superior to that of rTMS studies (*Gilio et al., 2003*) (Cohen's $d$=0.80; note that it is a read from the graph). Two meta-analyses of clinical trials using rTMS and tDCS also indicated that the effect sizes were medium (0.4–0.6) (*Adeyemo et al., 2012*; *Hsu et al., 2012*). Based on a previous study of neurofeedback training combined with externally administered interventions (*Ang et al., 2015*), their combination may contribute to facilitating neural plasticity.

Our techniques and evidences are expected to be applied in various fields, e.g., in the context of neurorehabilitation. In stroke patients, the interhemispheric imbalance/competition model predicts the presence of asymmetry in the interhemispheric sensorimotor network, with excessive inhibition from the non-affected hemisphere limiting motor performance and maximal recovery (*Duque et al., 2005*; *Murase et al., 2004*). Therefore, it is believed that guiding inhibitory interhemispheric network to the appropriate pattern through both targeted up-conditioning in the affected hemisphere and down-conditioning in the non-affected hemisphere may contribute to reduced abnormal IHI and enhanced achievable functional recovery (*Chieffo et al., 2013*; *Di Pino et al., 2014*; *Dong et al., 2006*; *Hummel and Cohen, 2006*). Although the rehabilitation strategy of attempting to rebalance interhemispheric networks in order to improve motor recovery after stroke is controversial (*Bundy et al., 2017*; *Carson, 2020*; *Xu et al., 2019*), the current technique can be tailored either to up-conditioning the damaged hemisphere, down-conditioning the intact hemisphere, a combination of both, or vice versa depending on the patient's specific states. Furthermore, a previous review advocated the conceptual framework and suggested that a fundamental role of IHI is to support the narrowing of excitatory focus by co-opting the capacities of the two cerebral hemispheres (*Carson, 2020*). This framework is consistent with the known associations between the structural integrity of callosal projections and the magnitude of the motor deficits after stroke (*Auriat et al., 2015*; *Granziera et al., 2012*; *Koh et al., 2018*). Thus, our spatially bivariate EEG-neurofeedback approach might be highly relevant to application perspective in the context of stroke recovery as well as a basic science perspective. Future work probing the residual ability to manipulate IHI in stroke patients is warranted.

## Limitations

Our study has several limitations that need to be considered when interpreting the results. Although we successfully demonstrated the bidirectional changes in the ipsilateral SMR-ERD, the bidirectional changes in IHI from the right to the left hemisphere were not statistically significant since there was

no significant difference between the MID and LOW sessions (*Figure 4B*). The current results should prompt future studies to ascertain whether down-regulation of IHI (i.e. disinhibition) during MI and up-regulation of IHI at rest in the non-dominant hemisphere (intact hemisphere) are possible in a similar setup.

Additionally, a methodological limitation of our study was related to the order effects due to cumulative TMS pulses and elapsed time since the order of REST and NoFB sessions were not randomized (the last three neurofeedback sessions [HIGH, MID, and LOW] were randomized). However, statistical analyses included all sessions, including REST and NoFB sessions, using the latter two sessions as a control condition. Therefore, our comparison to REST and/or NoFB sessions may have been affected by the order effect.

For the limitations of the volitional control of SMR-ERD during unilateral MI, all participants were right-handed (Laterality Quotient: 72.2 ± 30.9%), as assessed by the Edinburgh Inventory (*Oldfield, 1971*), to eliminate bias due to the dominance. If a left-handed participant performs the same experiment, the modulation effect via BCI-based neurofeedback may be different. Moreover, in case the contralateral SMR-ERD was up- or down-regulated instead of being maintained at a moderate level, it is unclear whether the ipsilateral SMR-ERD was modulated bidirectionally and whether the IHI changed with it. The study performed a total of 11 sessions, including REST and NoFB, and the combinations of SMR-ERDs in both hemispheres (3 conditions versus 3 conditions) may be worth evaluating in the future. In addition, it is not clear from previous experiments whether the bivariate neurofeedback approach is superior to the conventional BCI-based neurofeedback that exploits SMR-ERD from one hemisphere. Since both hemispheres are connected by intrinsic transcallosal projections and exhibit functional crosstalk (*Arai et al., 2011*; *Hofer and Frahm, 2006*; *Meyer et al., 1995*; *Waters et al., 2017*), it is expected that the sensorimotor excitability in the opposite hemisphere would also be up-regulated through univariate (i.e. ipsilateral hemisphere to the imagined hand) BCI-based neurofeedback. Consistent with the findings that the unilateral up- or down-regulation of the sensorimotor excitability manipulates the IHI magnitude, such sensorimotor activation without spatial specificity may inhibit the other hemisphere, leading to weaker range of IHI manipulation. Thus, we presumed that it is difficult to produce the same effect as this study via univariate BCI-based neurofeedback. A future study is required to directly compare the effect size of IHI manipulation between bivariate and univariate BCI-based neurofeedback in a within-participant cross-over design with adjusting experimental time by omitting certain experimental conditions.

Finally, the sex ratio in this study was the consequence of recruiting freely from the university; there was no experimental plan to equalize the sex ratio. Because there were few female participants in the study, it is unclear whether our findings can be generalized to females. Previous studies revealed that the brain microstructure, including corpus callosum, differed between men and women (*Dunst et al., 2014*; *Menzler et al., 2011*). It is unknown whether these sex differences influenced our IHI study and additional studies are required to determine this.

## Conclusion

In conclusion, we presented an innovative approach to manipulate the state of IHI, by directly and bidirectionally modulating SMR-ERDs in a spatially bivariate BCI-based neurofeedback paradigm. This approach provides the opportunity to understand the inhibitory sensorimotor functions and paves the way for new technologies that allow the user/patient to regulate aspects of their brain function to reach the desired states, e.g., for neurorehabilitation and enhanced motor performance.

## Materials and methods
### Study design

The current study was performed in accordance with approved guidelines and regulations, such as the CONSORT Statement (*Moher et al., 2001*) and CRED-nf checklist (*Ros et al., 2020*). The experiment consisted of five sessions: (1) resting-state (REST), (2) right finger MI without visual feedback (NoFB), (3) high (HIGH), (4) middle (MID), and (5) low (LOW) excitability states of the ipsilateral SM1 during BCI-based neurofeedback (details in 'Experimental sessions'). The difference of IHI magnitude in the last three sessions (i.e. HIGH, MID, and LOW sessions) was the primary dependent variable of interest. The three neurofeedback sessions were conducted in a randomized order across participants. Prior

to the three sessions, REST and NoFB sessions were performed to estimate individual baseline during rest and MI for the offline analysis.

To estimate the appropriate sample size for this study, a preliminary experiment was conducted before the main experiment. In the preliminary experiment, four healthy participants (not included in the main experiment) performed BCI-based neurofeedback training and underwent brain state-dependent dual-coil brain stimulation, similar to the main experiment. We calculated the IHI magnitude in each session. Then, an a priori power analysis ($\alpha$=0.05, 1-$\beta$=0.8, two-sided tests, Bonferroni corrected) focusing on the IHI magnitude using the statistical package G*Power 3.1 (*Faul et al., 2009*) was conducted. Because the preliminary experiment showed a large effect size on the IHI differences between HIGH (65.0 ± 22.8, mean ± SD) and MID (90.7 ± 23.4) sessions (Cohen's $d$=1.12), and between MID (90.7 ± 23.4) and LOW (106.3 ± 13.1) sessions (Cohen's $d$=0.82), we calculated that 24 participants were needed (*Cohen, 1992*).

## Participants

About 24 volunteers (2 women and 22 men; mean age ± SD: 23.4 ± 2.0 years; age range: 21–27 years) participated in this study. All participants had normal or corrected-to-normal vision and reported no history of neurological or psychological disorders. All participants were right-handed (Laterality Quotient: 72.2 ± 30.9%) as assessed by the Edinburgh Inventory (*Oldfield, 1971*). The TMS procedure was not applied to two participants due to the higher RMT (greater than 70% of the maximum stimulator output) of the right or left FDI muscles. This criterion ensured that the TMS stimulator would be able to perform at the required intensities for the whole duration of the experiment (*Stefanou et al., 2018*; *Zrenner et al., 2018*). We did not exclude participants based on the EEG characteristics such as the magnitude of their endogenous SMR activity (*Madsen et al., 2019*; *Safeldt et al., 2017*), to verify our hypothesis in a proof-of-concept study. About 22 participants (2 women and 20 men; mean age ± SD: 23.3 ± 1.9 years; age range: 21–27 years; Laterality Quotient: 72.5 ± 32.2%) completed the IHI experiment. Therefore, the results presented in the EEG part are from all 24 participants, while IHI results are from the 22 participants that completed the whole experiment. Around 4 of the 120 sessions from 4 participants (2 REST, 1 NoFB, and 1 HIGH session) were excluded due to corrupted data.

The experiments conformed to the Declaration of Helsinki and were performed in accordance with the current TMS safety guidelines of the International Federation of Clinical Neurophysiology (*Rossi et al., 2009*). The experimental procedure was approved by the Ethics Committee of the Faculty of Science and Technology, Keio University (no.: 31–89, 2020–38, and 2021–74). Written informed consent was obtained from participants prior to the experiments.

## EEG/Electromyogram (EMG) data acquisition

The EEG signals were acquired using a 128-channel Hydrogel Geodesic Sensor Net 130 system (Electrical Geodesics Incorporated [EGI], Eugene, OR, USA) in a quiet room. The EEG data were collected at a sampling rate of 1 kHz and transmitted via an ethernet switch (Gigabit Web Smart Switch; Black Box, Pennsylvania, USA) to the EEG recording software (Net Station 5.2; EGI and MATLAB R2019a; The Mathworks, Inc, Massachusetts, USA). The ground and reference channels were placed at CPz and Cz, respectively. The impedance of all channels, excluding the outermost part, was maintained below 30 kΩ throughout the experiment to standardize the EEG recordings (*Ferree et al., 2001*). This impedance standard was consistent with other studies using the same EEG system (*Carter Leno et al., 2018*; *Robertson et al., 2019*).

Surface EMGs were recorded from the FDI and ADM of the left and right hands using two pairs of Ag/AgCl electrodes ($\phi$ = 10 mm) in a belly-tendon montage. Impedance for all channels was maintained below 20 kΩ throughout the experiment. The EMG signals were digitized at 10 kHz using Neuropack MEB-2306 (Nihon Kohden, Tokyo, Japan). The EMG data from each trial were stored for offline analysis on a computer from 500 ms before to 500 ms after the TMS pulse. Simultaneously, 5–10 ms of data were transferred immediately after collection to a computer for real-time analysis. In case of muscular contraction due to finger movement, the experimenter reminded participants to relax their muscles and ensure absence of muscle activity during MI. To monitor the real-time surface EMG signals, EMG signals were band-pass filtered (5–1000 Hz with second-order Butterworth) with a 50 Hz notch to avoid power-line noise contamination; the root mean square of the filtered EMG signal

from the FDI for the previous 1000 ms of data was displayed on the second experimenter's screen in the form of a bar.

Throughout the experiment, the participants were seated in a comfortable chair with stable forearm support and performed MI of unilateral right index finger abduction. The wrist and elbow joint angles were fixed to the armrest in a neutral posture. The participants were instructed to maintain this posture and were visually monitored by the experimenter throughout the EEG and MEP measurements. During MI, the forelimbs were placed in a prone position, with natural elbow and shoulder joint angles to prevent muscle activity.

## TMS protocol

For the evaluation of IHI, TMS was delivered using two interconnected single-pulse magnetic stimulators (The Magstim BiStim$^2$; Magstim, Whiteland, UK) producing two monophasic current waveforms in a 70 mm figure-of-eight coil. We identified the optimal left and right coil positions over the hand representation area at which a single-pulse TMS evoked a MEP response in the FDI muscle with the lowest stimulus intensity, referred to as the motor hotspot. The TS was delivered to the motor hotspot of the left M1, with the handle of the coil pointing backward and approximately 45° to the midsagittal line. The other coil for the CS was placed over the motor hotspot of the right M1 but slightly reoriented at 45–60° relative to the midsagittal line because it was not possible to place two coils in some participants with small head size. This orientation is often chosen in IHI studies (*Daskalakis et al., 2002*) since it induces a posterior-anterior current flow approximately perpendicular to the anterior wall of the central sulcus, which evokes MEPs at the lowest stimulus intensities (*Rossini et al., 2015*). To immobilize the head and maintain fixed coil positions over the motor hotspots during the experiment, chin support and coil fixation arms were used. The position of the TMS coil was monitored using the Brainsight TMS navigation system (Rogue Research, Cardiff, UK), so that the optimal coil orientation and location remained constant throughout the experiment.

RMT was defined as the lowest stimulator output eliciting a MEP in the contralateral side of relaxed FDI of >50 μV peak-to-peak in 5 out of 10 consecutive trials (*Groppa et al., 2012*; *Rossini et al., 1994*). The stimulus intensities of the left and right M1 to evoke MEP of 1 mV peak-to-peak amplitude from the relaxed right and left FDIs ($SI_{1mV}$) were also determined for the following dual-coil paired-pulse TMS.

## IHI evaluation

The IHI from the ipsilateral (right) to the contralateral (left) M1 was probed using a dual-coil paired-pulse TMS paradigm. The CS was applied to the right M1, followed a few milliseconds later by a TS delivered to the left M1 (*Ferbert et al., 1992*). Due to the time constraint, the inter-stimulus interval (ISI) in the present study was uniformly set to 10 ms, in accordance with previous studies (*Duque et al., 2005*; *Harris-Love et al., 2007*; *Murase et al., 2004*; *Tsutsumi et al., 2012*); however, an ideal ISI would vary across individuals. Additionally, in a preliminary experiment with four participants, it was confirmed that IHI was clearly observed when ISI was set to 10 ms. The CS and TS intensities remained constant throughout the experiment for each participant.

To validate IHI measurement under bi-EEG-triggered dual-TMS setup, IHI curves at rest were obtained in 20 of 24 participants prior to the main experiment, where a CS of varying intensity (five different intensities, 100–140% of RMT, in steps of 10% RMT) preceded the TS. Around 10 conditioned MEPs were collected for each CS intensity, along with 10 unconditioned MEPs (i.e. TS was given alone) in randomized order. The peak-to-peak amplitudes of the conditioned MEPs were averaged for the different CS intensities and expressed as a percentage of the mean unconditioned MEP amplitude. The IHI intensity curves (*Figure 3*) ensured that IHI was approximately half-maximum for each participant when 120–130% RMT of CS intensity was applied, similar to the previous EEG-TMS experiment (*Stefanou et al., 2018*; *Tsutsumi et al., 2012*).

## Spatially bivariate BCI-based neurofeedback

The present study was conducted based on a spatially bivariate BCI-based neurofeedback that displays bi-hemispheric sensorimotor cortical activities, which we recently developed in our laboratory (*Hayashi et al., 2021*; *Hayashi et al., 2020*). This method allows participants to learn to regulate these two variates at the same time and induce changes in target-hemisphere-specific SMR-ERD. Visual feedback



**Video 1.** Sample video of closed-loop bivariate (bi-hemispheric brain state-dependent) electroencephalography (EEG)-triggered dual-coil transcranial magnetic stimulation (TMS) system (bi-EEG-triggered dual-TMS).
https://elifesciences.org/articles/76411/figures#video1

was provided on a computer screen in the form of cursor movements in a two-dimensional coordinate, in which x- and y-axis corresponded to the degree of the ipsilateral (right) and contralateral (left) SMR-ERD to the imagined right hand, respectively. The axis range was set from the 5th (i.e. SMR-ERS) to 95th (i.e. SMR-ERD) percentile of intrinsic SMR-ERD distribution in the EEG calibration session, and the origin-position (x=0, y=0) represented median values of bilateral SMR-ERDs, respectively. The cursors were presented at the origin-position at the initiation of a trial, and values exceeding the boundary were rounded to either the 5th or 95th percentile. A key point of the current methodology is that, e.g., when participants were instructed to move the cursor toward the middle right (x>0, y=0) in the two-dimensional coordinate, the position showed a strong SMR-ERD in the ipsilateral hemisphere and moderate SMR-ERD in the contralateral hemisphere. Therefore, spatially bivariate BCI-based neurofeedback enables us to investigate how sensorimotor excitability in the target hemisphere (i.e. ipsilateral side to the imagined hand) contributes to IHI from the ipsilateral hemisphere to the contralateral hemisphere while maintaining constant contralateral sensorimotor excitability.

During MI, participants were asked to perform kinesthetic MI of right index finger abduction from the first-person perspective with equal time constants of 0.5 Hz cycle. Kinesthetic MI was performed because a previous study demonstrated that the focus of EEG activity during kinesthetic MI was close to the sensorimotor hand area, whereas visual MI did not reveal a clear spatial pattern (*Neuper et al., 2005*). To improve MI task compliance (i.e. whether all participants successfully performed the MI in the same manner), we not only asked them to perform kinesthetic MI from a first-person perspective but also asked them to perform a rehearsal before each session. In addition, we confirmed that SMR was observed in a frequency-specific, spatiotemporal-specific, and MI-related manner through offline analysis after each session. These characteristics of SMR-ERD indicate that kinesthetic MI, not visual MI, was performed appropriately (*Neuper et al., 2005*; *Pfurtscheller and Neuper, 1997*).

## Real-time brain state-dependent dual-coil brain stimulation

The bi-EEG-triggered dual-TMS setup analyzes the raw EEG signal in real time to trigger TMS pulses depending on the instantaneous bilateral spatially filtered SMR-ERDs (see *Video 1*). The real-time SMR-ERD intensity in each hemisphere (relative to the average power of the 1–5 s of the resting epoch) was obtained every 100 ms and calculated using the last 1 s data as follows *Hayashi et al., 2020*: (1) acquired raw EEG signals recorded over SM1 underwent a 1–70 Hz second-order Butterworth band-pass filter and a 50 Hz notch filter; (2) filtered EEG signals were spatially filtered with a large Laplacian (60 mm to set of surrounding channels), which subtracted the average value of the surrounding six channel montage from that of the channel of interest (i.e. C3 and C4, respectively). This method enabled us to extract the task-related EEG signature and improve the signal-to-noise ratio of SMR signals (*McFarland et al., 1997*; *Tsuchimoto et al., 2021*). In addition, the large Laplacian method is better matched to the topographical extent of the EEG control signal than the small Laplacian and ear reference methods *McFarland et al., 1997*; (3) a fast Fourier transform was applied to the spatially large Laplacian filtered EEG signals; (4) the power spectrum was calculated by calculating the square of the Fourier spectrum; (5) the alpha band power was obtained by averaging the power spectrum across the predefined alpha target frequencies from the EEG calibration session (described below); (6) the alpha band power was time-smoothed by averaging across the last five windows (i.e. 500 ms) to extract the low-frequency component for high controllability. The low-frequency component is beneficial to neurofeedback training because it reduced the flickering and improved the signal-to-noise ratio of the SMR signal *He et al., 2020*; *Kober et al., 2018*; and (7) SMR-ERD was obtained

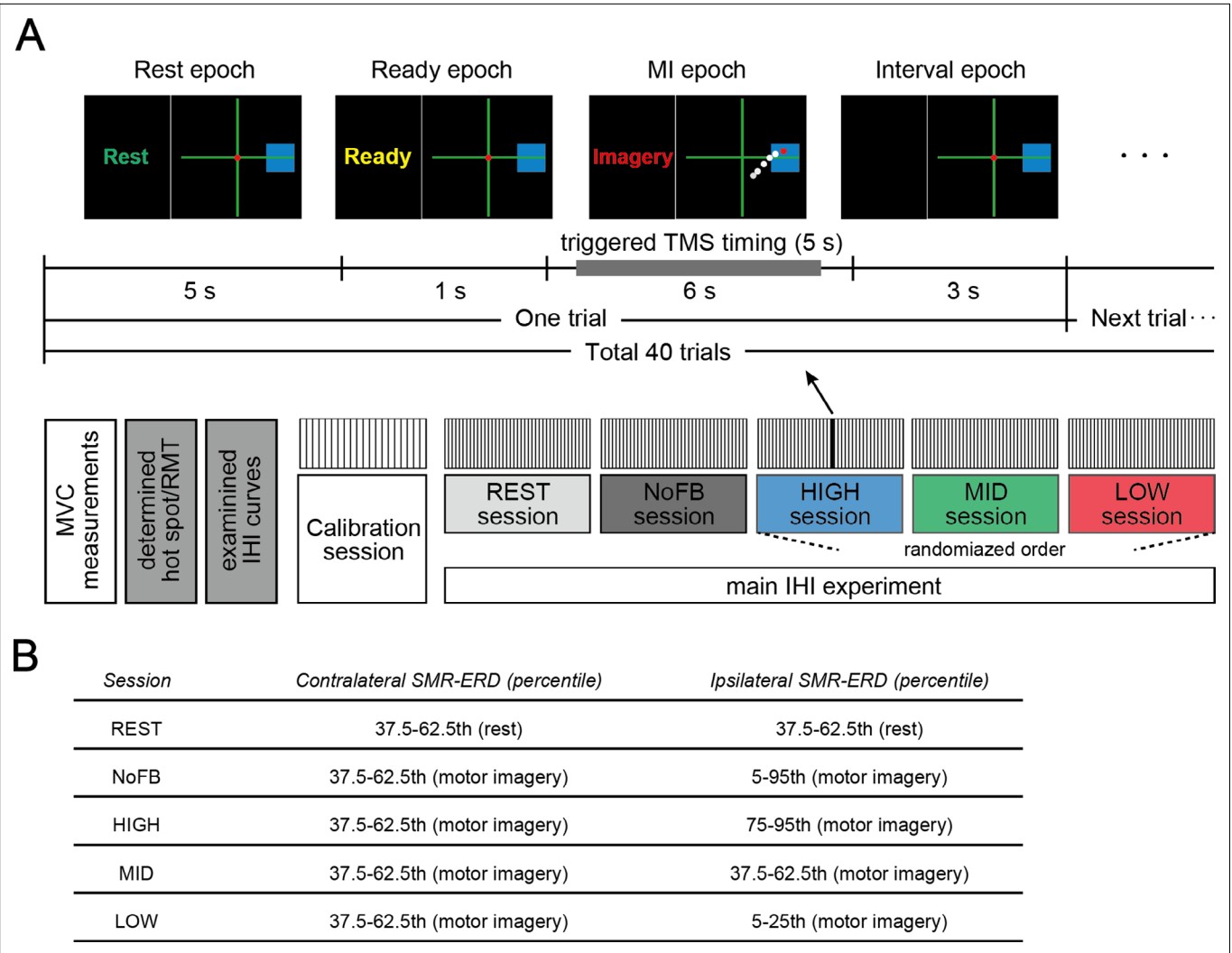

**Figure 7.** Experimental paradigm. (**A**) Task instructions and visual SMR-ERD feedback in the contralateral and ipsilateral sensorimotor cortex (SM1) were provided in the form of computer cursors in a two-dimensional coordinate on a computer screen (upper panel). The electroencephalography (EEG)-triggered transcranial magnetic stimulation (TMS) timing was determined based on the intrinsic sensorimotor cortical activity of each participant in the calibration session and ranged from 0.5 to 5.5 s during the motor imagery (MI) epoch. We also applied non-triggered TMS if SMR-ERD was not achieved with the target ranges (referred to as the failed trial). In the non-triggered TMS trials, paired pulses or unconditioned test pulses were delivered in random timing ranging from 5.5 to 6 s irrespective of instantaneous SMR-ERD during the MI epoch to see the influence of spontaneous SMR fluctuations on interhemispheric inhibition (IHI). Lower panel indicates the experimental overview. The last three HIGH, MID, and LOW sessions were arranged in a random order, and in these sessions, participants received visual feedback based on bilateral SMR-ERDs. (**B**) The predetermined target ranges of SMR-ERD were expressed by a blue rectangle on the computer screen in each session. We aimed for participants to volitionally increase or decrease (bidirectional) the ipsilateral sensorimotor excitability while maintaining constant contralateral sensorimotor excitability. The REST and NoFB sessions were served in order to estimate the individual baseline during rest and MI for the offline analysis.

by calculating the relative power to the average power during the resting epoch. The SMR-ERDs of the last six segments were displayed and each plot was updated every 100 ms, respectively, allowing participants to always see the SMR-ERD from 0.6 s ago to the present. Thereafter, a cue signal was generated to trigger the magnetic stimulator of CS stimulus when the signal reached the predetermined target that the SMR-ERD threshold was exceeded and transmitted transistor-transistor logic pulse to the magnetic stimulator of TS stimulus 10 ms later by Neuropack MEB-2306 system. EMG and EEG data were processed using the customized analysis scripts on MATLAB R2019a (*Hayashi, 2022*).

## Experimental sessions

First, maximal voluntary contraction (MVC) was measured (*Figure 7A*). Full-length isometric abduction of the right and left index and little fingers were performed once after several exercises; each execution lasted 5 s with a 30 s rest between contractions to allow for recovery from mental fatigue. Each MVC was obtained by calculating the root mean square of stable 3 s of filtered EMG data. Then, hot spots and stimulus intensities were determined.

Next, to determine the parameters of the bi-EEG-triggered dual-TMS setup, an EEG calibration session consisting of 20 trials, providing real-time SMR-ERD only on the contralateral side, was performed for each participant prior to the IHI experiment. Each trial was initiated by a 5 s resting epoch, followed by a 1 s ready epoch, and completed by a 6 s MI epoch. During this 12 s trial period, participants were asked not to move, blink, or swallow to prevent EEG artifacts derived from non-neural activity. After each 12 s trial, the screen went black for 3 s (*Figure 7A*). Participants were allowed to move freely to avoid mental fatigue during this interval period, before the next trial started. Thereafter, the target frequencies in the contralateral and ipsilateral SM1 were determined for each participant in order to feedback the most reactive frequency. Since SMR-ERD in the alpha band is a reliable EEG biomarker of increased neuronal excitability in SM1, corticospinal tract, and thalamocortical systems (*Neuper et al., 2006*; *Soekadar et al., 2015b*; *Takemi et al., 2018*; *Takemi et al., 2015*; *Takemi et al., 2013*; *Yuan et al., 2010*), the target frequencies were selected from the alpha band (8–13 Hz) by calculating the mean intensity of SMR-ERD with a 3 Hz sliding bin and 2 Hz overlap. Second, the target ranges of SMR-ERD during bi-EEG-triggered dual-TMS setting were normalized for each participant based on SMR-ERD distribution in the contralateral and ipsilateral hemispheres.

After the calibration session, the main IHI experiment with dual-coil paired-pulse TMS was performed in five consecutive sessions (10 min each) with fixed CS and TS intensities. Each session consisted of 12 s of 40 trials with 3 s interval periods as same as the EEG calibration session. Five experimental sessions comprised different conditions as follows: (1) resting-state where participants were instructed to relax and look at the origin of the 2-D coordinates on the computer screen in front of them (REST), (2) right finger MI without visual feedback (NoFB), participants tried to achieve (3) high (HIGH), (4) middle (MID) and (5) low (LOW) excitability states of the ipsilateral SM1 during BCI-base neurofeedback. In the last three HIGH, MID, and LOW sessions, participants received visual feedback based on the SMR-ERDs from both contralateral and ipsilateral hemispheres.

In each session, approximately equal numbers of paired pulses and unconditioned test pulses were applied. EEG-triggered TMS timing was determined based on the intrinsic sensorimotor cortical activity of each participant in the calibration session and ranged from 0.5 to 5.5 s during the MI epoch. The predetermined target ranges of SMR-ERD was expressed by a blue rectangle on the computer screen in each session are as follows (*Figure 7B*): (1) 37.5–62.5th percentile of SMR-ERD distribution during rest in both hemispheres (REST session); (2) 37.5–62.5th percentile of SMR-ERD distribution during MI in the contralateral hemisphere and 5–95th percentile of SMR-ERD distribution during MI in the ipsilateral hemisphere (NoFB session); (3) 37.5–62.5th percentile of SMR-ERD distribution during MI in the contralateral hemisphere and 75–95th percentile of SMR-ERD distribution during MI in the ipsilateral hemisphere (HIGH session); (4) 37.5–62.5th percentile of SMR-ERD distribution during MI in the contralateral hemisphere and 37.5–62.5th percentile of SMR-ERD distribution during MI in the ipsilateral hemisphere (MID session); and (5) 37.5–62.5th percentile of SMR-ERD distribution during MI in the contralateral hemisphere and 5–25th percentile of SMR-ERD distribution during MI in the ipsilateral hemisphere (LOW session).

We also applied non-triggered TMS if SMR-ERD was not achieved with the target ranges (referred to as the failed trial). In the non-triggered TMS trials, paired pulses or unconditioned test pulses were delivered in random timing ranging from 5.5 to 6 s irrespective of instantaneous SMR-ERD during the MI epoch to see the influence of spontaneous SMR fluctuations on IHI. The REST and NoFB sessions served as controls to determine the intrinsic IHI magnitude. To evaluate the difficulty of each neurofeedback session, the mean values of the success triggered trials (±1 SD) in all participants were compared. The waiting time for a triggered event from MI onset was also measured for each session.

## MEP analysis

For the quality control of MEP analysis, trials were rejected if: (1) coil position was shifted from the optimal orientation and location (>3 mm and/or >3°) despite maintaining it during the experiment

using the Brainsight TMS navigation system; (2) involuntary muscle contraction in the 250 ms period before the TMS pulse was observed (>5% MVC) because of pre-innervation increase in MEP amplitude (*Devanne et al., 1997*; *Hallett, 2007*); (3) large trial-by-trial MEP variance (mean ± 3 SD) were found in order to screen out extreme values (*Ruddy et al., 2018*). In total, 8.8% of all trials were excluded from further analysis. Each peak-to-peak MEP amplitude was automatically determined in the remaining trials within 20–45 ms after the TMS pulse. The IHI was defined as the percentage of mean conditioned MEP amplitude over mean unconditioned MEP amplitude (IHI = conditioned MEP/ unconditioned MEP × 100%); therefore, smaller IHI values represent stronger inhibition.

## Offline EEG analysis

To evaluate the sensorimotor excitability that may influence IHI, pre-processing and time-frequency analyses were performed, and the left and right SMR-ERDs were calculated. The SMR-ERD in EEG is a reliable surrogate monitoring marker of sensorimotor excitability level for several reasons: (1) SMR-ERD and task-induced increase in blood oxygenation level-dependent signals during MI are co-localized and co-varied at SM1 *Yuan et al., 2010*; (2) SMR-ERD control is associated with the contribution of SM1 modulated by tDCS (*Soekadar et al., 2015b*); and (3) data-driven EEG features discriminating the presence or absence of muscle contraction were predominantly localized in the parieto-temporal regions, indicating SMR-ERD (*Hayashi et al., 2019*; *Iwama et al., 2020*). The time segment of interest was from the initiation of the trial to before the TMS-triggered time marker of the CS in order to avoid contamination by the TMS artifact (pre-stimulation period). The EEG signal underwent a 1–70 Hz, second-order Butterworth bandpass filter and a 50 Hz notch filter. The EEG signals of all channels were spatially filtered using a common average reference, which subtracted the average value of the entire electrode montage (the common average) from that of the channel of interest to remove global noise (*McFarland et al., 1997*; *Tsuchimoto et al., 2021*). EEG channels in each trial were rejected during further analysis if they contained an amplitude above 100 μV (*Sanei and Chambers, 2013*).

## Connectivity analysis

To assess interhemispheric functional connectivity at the EEG level, distributed interregional neural communication was calculated. We focused on both the resting epoch (1–5 s) and MI epoch (7 s [task onset] before stimulation) for analysis. To calculate functional connectivity and compensate for long-range synchronization preference, we used the ciCOH (*Ewald et al., 2012*; *Hayashi et al., 2020*; *Vukelić and Gharabaghi, 2015*). The details of the following processing for the connectivity analysis can be obtained from our previous work (*Hayashi et al., 2020*). The ciCOH was obtained by subdividing the resting epoch into 1 s segments with 90% overlap (31 segments in total) and multiplied with a Hanning window. Then, interhemispheric Network-intensity was computed as follows:

$$Network\ intensity\ (f) = \sum\nolimits_{cont=1}^{7} \sum_{ipsi=1}^{7} ciCOH_{cont,ipsi}\ (f) \tag{2}$$

where *Network-intensity* is computed between left and right hemisphere, $\sum\sum ciCOH$ is the sum of significant ciCOH values for interhemispheric interaction, *cont* denotes the 7 channels of interest (i.e. C3 and its neighboring 6 channels), *ipsi* denotes the 7 channels of interest in the opposite hemisphere (i.e. C4 and its neighboring 6 channels), and *f* indicates the frequency of interest (i.e. the predefined alpha frequencies). As a negative control, other frequency bands (theta [4–7 Hz], low-beta [14–20 Hz], high-beta [21–30 Hz], and gamma [31–50 Hz]) were also examined.

## Correlation analysis

To investigate the association between sensorimotor brain activity at the EEG level and IHI magnitude from multimodal perspectives, within- and across-participant correlation analyses were performed. In the within-participant correlations between contralateral or ipsilateral SMR-ERDs and IHI magnitude in each participant, we used Pearson's correlation since MEP/EEG data showed normal distribution. In the across-participant corrections between contralateral or ipsilateral SMR-ERDs and IHI magnitude, we performed a repeated measures correlation. A previous study argued that it may produce biased, specious results due to violation of independence and/or differing patterns (across-participants versus within-participants) (*Bakdash and Marusich, 2017*).

To examine the neural characteristics depending on the manipulation capability of IHI, we examined the association between the manipulated effects on IHI calculated from the percentage change between HIGH and LOW sessions ($\Delta IHI_{H-L}$), and IHI magnitude in REST session ($IHI_{rest}$). Next, the correlations between $IHI_{rest}$ and intrinsic EEG profile in NoFB session (i.e. contralateral SMR-ERD and ipsilateral SMR-ERD) were investigated, respectively. Moreover, we verified whether large-scale resting-state functional connectivity was associated with an effective inhibitory interhemispheric network assessed by IHI. An across-participant Pearson's correlation was applied to identify significant relationships between the $IHI_{rest}$ and interhemispheric Network-intensity$_{rest}$. To attenuate the aberrant effect of values in some especially low Network-intensity$_{rest}$ or high (CS + TS)/TS at rest (outliers) in the theta, low-beta, high-beta, and gamma bands, a jackknife resampling was performed using SPSS software (version 27; IBM Corp., Armonk, NY, USA). In this study, both of two correlation coefficients and the bias were reported.

## Statistical analysis

Statistical analyses were performed using SPSS software, MATLAB R2019a, and R 4.1.1 software. The assumption of normality was verified using the Shapiro-Wilk test. All data were normally distributed (W>0.91, p>0.05) and therefore analyzed with parametric tests. The assumption of sphericity was checked using the Mauchly's test. If the test was significant, a Greenhouse-Geisser correction was applied.

Modulation effects of bilateral sensorimotor excitabilities due to BCI-based neurofeedback training were evaluated from the 1 s period immediately before stimulation onset. A one-way rmANOVA for sessions (five levels: REST, NoFB, HIGH, MID, and LOW) was performed using the contralateral and ipsilateral SMR-ERDs. Following the one-way rmANOVA, post-hoc two-tailed paired t-tests were performed using the Bonferroni correction for multiple comparisons. For the interhemispheric connectivity during MI, a one-way rmANOVA and post-hoc analysis were applied to Network-intensity$_{MI}$ as the same procedure as the SMR-ERD analysis. In addition, to verify the difficulties for neurofeedback sessions, one-way rmANOVA for sessions (HIGH, MID, and LOW) and post-hoc analysis were applied to the number of the triggered (success) trials and mean waiting time for a triggered event from MI onset.

For the IHI curves at rest, a one-way rmANOVA for intensities (six levels: 0 [TS-only], 100, 110, 120, 130, and 140% RMT) and post-hoc two-tailed paired t-tests were performed in MEP amplitude. Similarly, a one-way rmANOVA for sessions (five levels: REST, NoFB, HIGH, MID, and LOW) was performed to compare the IHI magnitude. Next, for all significant main effects, post-hoc two-tailed paired t-tests were performed using the Bonferroni correction for multiple comparisons for all sessions.

We compared the IHI magnitude across sessions, by calculating the percentage changes in IHI magnitude to baseline (i.e. NoFB session) and investigating the difference between sessions (REST, HIGH, MID, and LOW). To investigate the influence of spontaneous SMR fluctuations on IHI, a two-way rmANOVA for sessions (HIGH, MID, and LOW) and trials (triggered TMS trial and non-triggered TMS trial) as the within-participant factors. To investigate whether changes in IHI and corticospinal excitability in response to the CS over the right hemisphere are independent phenomena, a generalized linear model with the percentage changes of left MEP amplitude as a covariate of no interest conducted for the triggered and non-triggered TMS trials. Following the two-way rmANOVA, a post-hoc analysis was performed using the Bonferroni correction for multiple comparisons. The significance level for all statistical tests was set to p=0.05. Although analysis of mean IHI magnitude across participants revealed significant modulation of the ipsilateral SMR-ERD (*Figure 2B*), we further performed mixed-effects analysis (*Hussain et al., 2019*; *Madsen et al., 2019*) incorporating the contralateral SMR-ERD and ipsilateral SMR-ERD as factors in the statistical regression model to explore the relationships between inhibitory interhemispheric activity and EEG characteristics. The linear mixed-effect model included the contralateral and ipsilateral SMR-ERD as fixed effects, treating the participant factor as a random effect to account for individual variability in IHI magnitude.

For the within-subject correlation analysis, the overall significance of all individual correlations between IHI magnitude and bilateral SMR-ERDs was examined using a chi-squared test.

## Acknowledgements

The authors would like to thank Kohsuke Okada for experimental assistance and to appreciate Sayoko Ishii, Kumi Nanjo, Yoko Mori, Yumiko Kakubari, Shoko Tonomoto, and Aya Kamiya for their technical support during the study. Funding for the development of the BCI system was supported by the Moonshot R&D program (Grant Number JPMJMS2012) from Japan Science and Technology Agency (JST) to J Ushiba. The physiological experiment was supported by a Grant-in-Aid for Transformative Research Areas (A) (#20H05923) from the Ministry of Education, Culture, Sports, Science and Technology (MEXT) and Strategic International Brain Science Research Promotion Program (Brain/MINDS Beyond) (#JP20dm0307022) from the Japan Agency for Medical Research and Development (AMED) to J Ushiba. In addition, this study was also supported by The Keio University Doctorate Student Grant-in-Aid Program from Ushioda Memorial Fund to M Hayashi.

## Additional information

### Competing interests

Masaaki Hayashi: is employed by LIFESCAPES Inc. Junichi Ushiba: is a founder and the Representative Director of the University Startup Company, LIFESCAPES Inc. involved in the research, development, and sales of rehabilitation devices including brain-computer interfaces. He receives a salary from Connect Inc., and holds shares in Connect Inc. This company does not have any relationships with the device or setup used in the current study. The other authors declare that no competing interests exist.

### Funding

| Funder | Grant reference number | Author |
| --- | --- | --- |
| Ministry of Education, Culture, Sports, Science and Technology | Grant-in-Aid for Transformative Research Areas (A) (#20H05923) | Junichi Ushiba |
| Japan Agency for Medical Research and Development | Strategic International Brain Science Research Promotion Program (#JP20dm030702) | Junichi Ushiba |
| Japan Science and Technology Agency | Moonshot R&D program (Grant Number JPMJMS2012) | Junichi Ushiba |
| Ushioda Memorial Fund | The Keio University Doctorate Student Grant-in-Aid Program | Masaaki Hayashi |

The funders had no role in study design, data collection and interpretation, or the decision to submit the work for publication.

### Author contributions

Masaaki Hayashi, Conceptualization, Data curation, Software, Formal analysis, Funding acquisition, Validation, Investigation, Visualization, Methodology, Writing – original draft, Project administration, Writing – review and editing; Kohei Okuyama, Data curation, Validation, Investigation, Methodology, Writing – review and editing; Nobuaki Mizuguchi, Validation, Investigation, Methodology, Writing – review and editing; Ryotaro Hirose, Taisuke Okamoto, Methodology, Writing – review and editing; Michiyuki Kawakami, Resources, Validation, Writing – review and editing; Junichi Ushiba, Conceptualization, Resources, Supervision, Funding acquisition, Methodology, Writing – review and editing

### Author ORCIDs

Masaaki Hayashi ⓘ http://orcid.org/0000-0002-7104-6765
Junichi Ushiba ⓘ http://orcid.org/0000-0003-1161-983X

### Ethics

The experiments conformed to the Declaration of Helsinki and were performed in accordance with the current TMS safety guidelines of the International Federation of Clinical Neurophysiology (Rossi et al., 2009). The experimental procedure was approved by the Ethics Committee of the Faculty of

Science and Technology, Keio University (no.: 31-89, 2020-38, and 2021-74). Written informed consent was obtained from participants prior to the experiments.

### Decision letter and Author response
Decision letter https://doi.org/10.7554/eLife.76411.sa1
Author response https://doi.org/10.7554/eLife.76411.sa2

## Additional files

### Supplementary files
• Supplementary file 1. Values corresponding to each statistical figure.
• Transparent reporting form

### Data availability
Source data used to generate the figures are publicly available via Dryad Digital Repository, accessible here: Hayashi, Masaaki (2021), Spatially bivariate EEG-neurofeedback can manipulate interhemispheric rebalancing of M1 excitability, Dryad, Dataset, https://doi.org/10.5061/dryad.hhmgqnkj3 Scripts used for the neurofeedback experiment are available on GitHub (https://github.com/MasaakiHayashi/elife-neurofeedback-experiment, (copy archived at swh:1:rev:20fab186058b4f59577c1d4da31f9fff377a129e)).

The following datasets were generated:

| Author(s) | Year | Dataset title | Dataset URL | Database and Identifier |
|---|---|---|---|---|
| Hayashi M | 2021 | Data from Spatially bivariate EEG-neurofeedback can manipulate interhemispheric rebalancing of M1 excitability | https://doi.org/10.5061/dryad.hhmgqnkj3 | Dryad Digital Repository, 10.5061/dryad.hhmgqnkj3 |
| Hayashi M | 2022 | Spatially bivariate EEG-neurofeedback can manipulate interhemispheric inhibition | https://github.com/MasaakiHayashi/elife-neurofeedback-experiment | GitHub, elife-neurofeedback-experiment |

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
