## [Editor Report]

The authors developed a new method for self-regulating interhemispheric inhibition via a novel brain-computer interface. The effect on interhemispheric inhibition was observed over and above modulating cortical excitability of a single hemisphere.

---

## [Decision Letter]

**Decision letter after peer review:**

Thank you for submitting your article "Spatially bivariate EEG-neurofeedback can manipulate interhemispheric rebalancing of M1 excitability" for consideration by *eLife*. Your article has been reviewed by 3 peer reviewers, including Nicole Wenderoth as the Reviewing Editor and Reviewer #1, and the evaluation has been overseen by and Richard Ivry as the Senior Editor. The following individual involved in the review of your submission has agreed to reveal their identity: Tomas Ros (Reviewer #2).

Essential revisions:

1) The main claim of the authors that the spatially bivariate EEG-neurofeedback lead to changes in interhemispheric inhibition is currently not sufficiently supported by the experimental evidence because (i) the authors did not demonstrate that IHI could be modulated bidirectionally, (ii) the training did not lead to a corresponding change in interhemispheric coherence and, therefore, no conclusions with regard to interhemispheric "rebalancing" can be drawn, and (iii) it is unclear whether the effects were mainly driven by excitability changes in the right hemisphere such that changes in IHI are merely an epiphenomenon. It will be important to address these concerns by presenting control analyses and/or additional data.

2) There is no significant difference in Network-intensity (better: interhemispheric coherence) during MI between HIGH and LOW sessions; the provided explanations in the discussion are highly speculative. The most obvious explanation is that interhemispheric coherence was simply not changed by MI and ERD feedback. How can this lack of an effect be reconciled with the main claim of the authors?

3) The paired-pulse TMS effects could result from modulating the excitability of the right motor cortex. It is important to show that IHI changed over and above the CS effect.

4) Bidirectional control of IHI would need to be significantly demonstrated above and below the resting baseline. Although the study shows bidirectional control of SMR power (in the ipsilateral hemisphere), this is strictly not the case for IHI (which shows only significant increases relative to rest).

5) The result section is difficult to understand because context information is often missing since the detailed Methods are reported at the end. Please revise the results to improve accessibility for the reader.

6) Some statements like "suggesting that the volitional control of SMR-ERDs in the closed-loop environment differs from spontaneous SMR fluctuations" require a direct statistical comparison

7) Revise correlation analyses to properly account for within versus between-subject variability

8) It remains an open question, whether the more sophisticated neurofeedback approach (of modulating both hemispheres separately) was indeed necessary to achieve the findings of this work.

9) The dynamic range of SMR modulation occurs mainly in the ipsilateral hemisphere (in terms of bidirectional control). The contralateral hemisphere consistently shows increased SMR power relative to rest, with no bidirectional modulation. Please explain why this was the case and discuss potential implications

10) Why was the gender ratio of the subjects (1:11, F:M) so highly skewed? This should be explained, and potential limitations should be discussed

11) The discussion of the neurophysiological mechanisms mediating IHI would benefit from more depth and a better embedding into the literature.

*Reviewer #1 (Recommendations for the authors):*

The manuscript reports the interesting phenomenon that selectively modulating the excitability of the right hemisphere changes interhemispheric inhibition from the right to the left hemisphere. While this is interesting, I'm wondering how the interhemispheric result relates to changes in corticospinal excitability as assessed via MEPs in response to the conditioning stimulus.

I understood that the CS has been chosen such that TS MEPs are reduced by approximately 50%. Was that done while participants were at rest or while they performed the MI task /BCI control?

If the CS intensity was determined at rest, did the authors analyse the CS MEPs and particularly how they change from IHI at rest to IHI during the MI/BCI control? Does IHI change independently of MEP modulation in response to the CS? This would be an important analysis also for the shuffled non-triggered trials since it might reveal whether corticospinal excitability and IHI can be controlled independently.

Please note that there has been some debate in the field whether or not the CS effect has to be kept constant across conditions. Thus, if there are differences in CS MEP amplitudes between the sessions, the authors might want to address this methodological issue in the discussion.

The result section is difficult to understand because context information is often missing since the detailed Methods are reported at the end. Please revise the results to improve accessibility for the reader. In that regard, I would recommend swapping the order in the result section and presenting first the EEG states and then the associated IHI measures.

What is the rationale for using right finger MI as the no-FB condition? Please explain

Line 257ff: there seems to be a mistake: «one-way rmANOVA for sessions (three levels: HIGH, MID, and LOW) revealed significant differences F(2,44) = 1.43, p < 0.248,…» should probably read « revealed no significant differences».

Line 260 ff there seems to be a mistake: Although the IHI magnitude in the LOW session was larger than that in the HIGH and LOW sessions.… replace second LOW by MID.

I liked the control analysis of the authors shown in suppl. Figure 3-2, However, the conclusion that the LOW,MID,HIGH data shown in Figure 3B is different from the data shown in suppl. Figure 3-2 requires a direct statistical comparison.

Figure 4C: I'm not convinced that presenting the laterality index adds any information to Figure 4 A and B. In fact, I found the caption confusing since it seems to suggest that (i) the negative LI for LOW results from contralateral activity which is inconsistent with the topoplot that shows high activity in the ipsilateral hemisphere; and (ii) the positive LI for HIGH results from ipsilateral activity which is also inconsistent with the topo which shows desynchronisation in the ipsilateral hemisphere. Given the shown in panels A and B, the LI measure seems inappropriate since there are virtually no activity changes in the contralateral hemisphere while the ipsilateral hemisphere shows either synchronized or desynchronized SMR activity.

Associations between IHI magnitude and bilateral EEG patterns: The authors might want to consider calculating a repeated measures correlation because it does not seem meaningful to pool data across the LOW, MID, HIGH sessions.

Discussion

Line 470ff: The first sentence of the discussion follows a reversed logic. Participants did not consciously self-regulate IHI but they modulated their EEG pattern based on the closed-loop spatially bivariate BCI-based neurofeedback. Changes in IHI were the consequence of self-regulating the EEG activity pattern. Please revise throughout the discussion.

The discussion of the neurophysiological mechanisms mediating IHI could be a bit more in-depth. The review of Carson RG. J Physiol. 2020 Nov;598(21):4781-4802 might help.

The authors argue that "This multimodal EEG-TMS approach proves that the bilateral α and β activities observed in the current study served to regulate the inhibition and facilitation of inhibitory functional coupling of interhemispheric interaction over motor cortices.» This argument would be strengthened if the interhemispheric effects were independent of the general excitability of the ipsilateral hemisphere as reflected in CS MEP amplitudes. Only then, the authors can argue that their intervention specifically modulated interhemispheric circuits.

Lines 538-544: Explanatory theoretical model: I cannot follow the argument of the authors. Is it possible that IHI and "network intensity" simply reflect the activity of different neural populations? IHI effects are rather specific and I doubt that this could be picked up via general EEG markers.

Line 561ff This seems to be consistent «Furthermore, the effect size of IHI manipulation in the current study (Cohen's d = 1.50) was approximately 1-2-fold higher than that of representative tDCS (Cohen's d = 1.55)…».

Discussion of IHI in stroke: As stated by the authors, it is somewhat controversial whether IHI is a useful therapy target early after stroke. However, their EEG-neurofeedback approach allows to selectively up- or down-regulate one hemisphere which might be highly relevant in the context of stroke recovery both from a basic science and application perspective. It might be worthwhile to expand this aspect of the discussion.

*Reviewer #2 (Recommendations for the authors):*

Although the Introduction is comprehensive in terms of the mechanistic relation between EEG and TMS variables, it does not sufficiently include the "clinical" significance of modifying IHI e.g. in terms of rehabilitation potential. It would be nice to include a few references in clinical populations where IHI and/or SMR are impacted, or where IHI/SMR have been used as prognostic markers of motor recovery. I would also include a sentence or two in the Abstract underlining the potential clinical significance of manipulating IHI.

In terms of related studies combining the TMS/EEG/BCI trio, the work by Ros et al. 2010 was one of the first to demonstrate that MEP measures of cortical excitability and inhibition (MEP, SICI) can be modulated using closed-loop feedback of oscillations from the motor cortex.

Ros, T., Munneke, M.A.M., Ruge, D., Gruzelier, J.H. and Rothwell, J.C. (2010), Endogenous control of waking brain rhythms induces neuroplasticity in humans. European Journal of Neuroscience, 31: 770-778. https://doi.org/10.1111/j.1460-9568.2010.07100.x

It is not clear what behavioural and neural variable the authors are referring to in the first Results section starting with. "We confirmed the difficulties of each session" (line 171).

Line 171: "The mean value of the sum of the triggered trials". Mean and sum have different definitions, do you mean 'the mean value of all the triggered trials'?

There is no reference to what the abbreviation 'TS' stands for (I am assuming it is "test stimulus")

Line 219: The manipulation range of IHI (i.e., the difference between HIGH and LOW sessions) was 32.6. However elsewhere the means (Line 222/ 223) are separately indicated as HIGH: 62.0 and LOW: 96.9. Their difference (96.9 – 62.0) is 34.9.

Line 263: "suggesting that the volitional control of SMR-ERDs in the closed-loop environment differs from spontaneous SMR fluctuations". This depends on how one defines the word "differs". In this case, the comparison is technically qualitative as the contrast is between a significant result (volitional control, p <0.05) and a non-significant one (spontaneous fluctuation, p > 0.05). However, a more direct comparison would involve comparing the two effects directly (volitional vs spontaneous) by statistically testing for a significant difference in effect sizes (e.g. using a two-way rmANOVA).

Taking the effect size of the HIGH-vs-LOW contrast as an example, for volitional control this is d = 1.42, while for spontaneous fluctuation it is d = 0.49, which yields a differential d = 0.93 (1.42 -0.49). This is likely statistically significant but it will depend on the error term, hence the need for a formal two-way rmANOVA analysis with an interaction effect. This should be done for all the contrast analyses in the paper (including SMR power), instead of using the one-way ANOVA.

The dynamic range of SMR modulation occurs mainly in the ipsilateral hemisphere (in terms of bidirectional control). The contralateral hemisphere consistently shows increased SMR power relative to rest, with no bidirectional modulation. How do the authors explain this ? Could this be due to how the neurofeedback was implemented for each hemisphere (thresholds/rewards)? Right hand dominance? If the latter, where there any differential effects in the subgroup of left hand dominant participants?

Line 580: "the current technique can be tailored either to upregulate the damaged hemisphere, down-regulate the intact hemisphere…" and Line 585: "In conclusion, we presented an innovative approach to voluntarily and bidirectionally manipulate the state of IHI"

Critically, bidirectional control would need to be significantly demonstrated above and below the resting baseline. Although the study shows bidirectional control of SMR power (in the ipsilateral hemisphere), this is strictly not the case for IHI (which shows only significant increases relative to rest). Hence, the term "bidirectional" should not be used in the context of the IHI, but rather the phrase "control of interhemispheric disinhibition". Hence, the authors could state that the current results should motivate future studies investigating whether upregulation of IHI in the damaged hemisphere is possible above and beyond spontaneous resting values.

*Reviewer #3 (Recommendations for the authors):*

This is a well-conducted study building and expanding on the pioneering work of this group (Takemi et al. 2013, 2015). The novel contribution of this study is that self-regulated ERD is not modulating the TMS effects on the periphery (MEP) only, but also the TMS effects on the opposite brain hemisphere. This has been probed by IHI, but may probably have also been shown by TEP.

All other interpretations with regard to "rebalancing", "pathway-specificity", "interhemispheric connectivity", and "challenging the concept of neurorehabilitation" etc. are far fetching and should be toned down. Such statements would have necessitated (i) either direct neurofeedback of interhemispheric coherence, (ii) or demonstration of a direct link between the self-regulated ERD and interhemispheric coherence, and (iii) the application in patient populations.

Therefore, it is advisable to better link this study to the large body of literature in the IHI field, and discuss where this work adds additional insights. It is surprising, for example, that relevant previous literature is not referred to and discussed: Liang et al. (2014) have shown that imagined movements change IHI, Nelson et al. (2009) demonstrated bi-directional IHI, and Morishita et al. (2011, 2012) contributed with regard to ipsilateral inhibitory circuits and IHI. This work should be cited.

The weaknesses listed above should also be acknowledged as limitations in the discussion.

[Editors' note: further revisions were suggested prior to acceptance, as described below.]

Thank you for resubmitting your work entitled "Spatially bivariate EEG-neurofeedback can manipulate interhemispheric inhibition" for further consideration by *eLife*. Your revised article has been evaluated by Richard Ivry (Senior Editor) and a Reviewing Editor.

The manuscript has been improved but there are some remaining issues that need to be addressed, as outlined below:

Based on the new analyses, it is unclear whether the original finding that interhemispheric inhibition can be modulated by a new EEG neurofeedback approach does still hold. Reviewer #1 points out an alternative statistical approach that might strengthen the original argument of the authors while controlling for changes in the excitability of the modulated hemisphere.

Reviewer #3 points out that it is still unclear whether bihemispheric EEG is superior to unihemispheric EEG. The authors might want to address this comment by adding an in-depth discussion of how their new neurofeedback method compares with existing methods with regard to modulating interhemispheric interactions.

*Reviewer #1 (Recommendations for the authors):*

The authors provided very thorough and extensive revisions and they have addressed most of my comments. It is particularly reassuring to see the additional analyses of corticomotor excitability as probed by the CS.

However, I'm still not completely convinced by the author's statistical approach.

If I understood the approach correctly, % change in IHI and % change of CS MEP amplitude were entered into one rmANOVA model and the authors tested for a condition x index interaction effect (which did not reach significance). This is a rather strict comparison. A more traditional approach would be to add % CS change as a covariate of no interest and test whether the IHI effect survives even if the covariate is added. If that is the case, the authors have demonstrated that IHI changes over and above the variance explained by the CS effect which would support their initial hypothesis that IHI can be modulated via their EEG-BCI and is not simply an epiphenomenon. No other statistical analyses would be required. However, the discussion would need to be modified!

The authors also analysed changes in CS MEP amplitude across conditions. CS MEPs did not significantly differ across the different EEG-BCI conditions. However, the absence of a significant effect does not allow the authors to conclude that there is "no effect". This would require a Bayesian approach which would test whether the data provides evidence in favour of H0. This could be another argument that IHI changes over and above CS changes. However, I don't think that these additional analyses would be needed if the co-variance analysis of IHI would reach significance. Maybe the authors might want to consult a stats expert for this analysis.

*Reviewer #3 (Recommendations for the authors):*

I am satisfied with the author's responses and revisions, which are extensive and much more consistent with the collected evidence from the experiment.

Ultimately, I am ambivalent (i.e. neither for nor against) on recommending this paper for publication in *eLife* based on its scientific significance. On a positive note, the authors clearly demonstrate that it is possible to modulate IHI (albeit unidirectionally) using bivariate neurofeedback, and this is an original and scientifically novel contribution. On the other hand, from the experiments conducted, it is not clear whether the bivariate neurofeedback approach is critical for the observed findings since there is no comparison to the conventional BCI-based neurofeedback that exploits SMR-ERD from one hemisphere. Hence the results show evidence of "sufficient" conditions to reduce inter-hemispheric inhibition (IHI) but do not provide evidence of "necessary" conditions i.e. whether uni-hemispheric neurofeedback would lead to similar effects but with a much more parsimonious experimental setup that does not involve controlling EEG from both hemispheres.

I, therefore, leave it to the senior editor for the ultimate decision regarding publication, taking also into account the other reviewers' comments.

---

## [Author Response]

Essential revisions:1) The main claim of the authors that the spatially bivariate EEG-neurofeedback lead to changes in interhemispheric inhibition is currently not sufficiently supported by the experimental evidence because (i) the authors did not demonstrate that IHI could be modulated bidirectionally, (ii) the training did not lead to a corresponding change in interhemispheric coherence and, therefore, no conclusions with regard to interhemispheric "rebalancing" can be drawn, and (iii) it is unclear whether the effects were mainly driven by excitability changes in the right hemisphere such that changes in IHI are merely an epiphenomenon. It will be important to address these concerns by presenting control analyses and/or additional data.

Overall, we agree with these opinions. We addressed all concerns corresponding as follows: (i) the response to No.4 below; (ii) the response to No.2 below; (iii) the response to No.3 below.

2) There is no significant difference in Network-intensity (better: interhemispheric coherence) during MI between HIGH and LOW sessions; the provided explanations in the discussion are highly speculative. The most obvious explanation is that interhemispheric coherence was simply not changed by MI and ERD feedback. How can this lack of an effect be reconciled with the main claim of the authors?

Our results presented a spatially bivariate EEG-based neurofeedback approach that enables healthy participants to independently self-regulate excitability of the left versus right hemispheres resulting in IHI changes. However, interhemispheric coherence was not associated with ipsilateral excitability (Figure 5D). We would like to emphasize that this exploratory analysis only provides a limited minor result.

We think that our results are reasonable since it is possible that interhemispheric coherence did not reflect inhibitory modulation via neurofeedback; IHI and Network-intensity may reflect the activity of different neural populations. Our explanatory theoretical model (Lines 538-544 in the previous manuscript) has not been fully clarified based on the experimental results, so we have removed the relevant text and simplified the discussion. It is unclear whether fMRI-based neurofeedback using interhemispheric functional connectivity is linked to ipsilateral excitability, and whether it modulates IHI magnitude.

In addition, the interhemispheric coherence was calculated with one-to-one electrodes, while the Network-intensity is an expanded index for evaluation between multiple electrodes. In this study, Network-intensity was adopted to evaluate coherence involving not only C3 (or C4) but also peripheral multiple electrodes. Previous studies described that Network-intensity was calculated from the corrected imaginary part of coherence (ciCOH) (Ewald et al., 2012; Hayashi et al., 2020; Vukelić and Gharabaghi, 2015). Interhemispheric Network-intensity was computed as follows (written in “Connectivity analysis”): Network intensity(f)= ∑cont=17∑ipsi=17ciCOHcont,ipsi(f) where *Network-intensity* is computed between left and right hemisphere, ∑∑*ciCOH* is the sum of significant ciCOH values for interhemispheric interaction, *cont* denotes the 7 channels of interest (i.e., C3 and its neighboring 6 channels), *ipsi* denotes the 7 channels of interest in the opposite hemisphere (i.e., C4 and its neighboring 6 channels), and *f* indicates the frequency of interest (e.g., the predefined α frequencies).

We have revised the related parts of the manuscript.

[Lines 858-860] “There was a significant difference in interhemispheric functional connectivity between HIGH and MID sessions, but not between HIGH and LOW sessions.”

[Lines 1142-1157] “However, even fMRI-based neurofeedback with high spatial resolution, based on blood oxygenation level-dependent signal, cannot easily distinguish between the excitatory and the inhibitory activities from each region (Moon et al., 2021). Additionally, neurofeedback using interhemispheric functional connectivity has the possibility to counteract the modulation of IHIs from the right to left hemisphere and from the left to right hemisphere. Our results showed no significant difference in Network-intensity during MI between HIGH and LOW sessions (Figure 5D), indicating that interhemispheric functional connectivity at the EEG level was not changed, whereas IHI was modulated. This may indicate that IHI and Network-intensity simply reflect the activity of different neural populations; interhemispheric projections act via surround/lateral inhibition in the sensory and motor cortices (Carson, 2020), while the synchrony at the EEG level reflects changes in the activity of local interactions between pyramidal neurons and interneurons in the thalamocortical loops (Pfurtscheller and Lopes da Silva, 1999). Furthermore, it is possible that Network-intensity reflects the degree of bidirectional synchrony between the two motor cortices, while IHI in this study reflects one-way (right-to-left) inhibitory effects.”

[Lines 1157-1159] “Therefore, it is unclear whether fMRI-based neurofeedback using interhemispheric functional connectivity is linked to ipsilateral excitability, and whether it modulates IHI magnitude.”

3) The paired-pulse TMS effects could result from modulating the excitability of the right motor cortex. It is important to show that IHI changed over and above the CS effect.

We appreciate your valuable comments. We have added additional results related to changes in corticospinal excitability as assessed by MEPs in the left FDI in response to the conditioning stimulus over the right hemisphere (Figure 4—figure supplements 3B). A direct statistical comparison between increase in IHI and left MEP amplitude is shown in Figure 4—figure supplements 3C. To compare the two indices, the percent changes based on the values in NoFB session were calculated. The stimulus intensity of the CS during the experiment was set such that the CS evoked MEP of peak-to-peak amplitude from the left FDIs (SI_1mV_) “at rest” (Lines 1482-1484, “The stimulus intensities of the left and right M1 to evoke MEP …”). In addition, we analyzed the non-triggered trials to show the CS effect (Figure 4—figure supplements 3D).

Based on the additional results, the MEP amplitudes in the left FDI elicited by CS did not vary across sessions, indicating that the modulation of the ipsilateral SMR-ERD caused no changes in corticospinal excitability in the contralateral side. Although we successfully revealed that modulating the excitability of the right hemisphere changes IHI from the right to the left hemisphere, a significant interaction between IHI modulation and changes in the left MEP amplitude was not observed. Thus, the specificity of the IHI manipulation could not be fully demonstrated. To confirm whether the participants learned pathway-specific IHI manipulation, the CS effect should have been kept constant across the conditions by adjusting the CS intensity to evoke left MEP of 1 mV peak-to-peak amplitude. Because such an experiment was not possible in the present study due to the time limitation (the experimental time would need to be doubled to 5–6 h to perform such an experiment), further studies are warranted.

We have toned down our results and conclusions, such as the use of the term "pathway-specific" manipulation, since our experiment did not directly demonstrate whether changes in interhemispheric inhibition and corticospinal excitability are independent phenomena. We have revised the manuscript to include these additional analyses, figures, and limitations.

[Lines 555-599] “Although we revealed that modulating the excitability of the right hemisphere changes IHI magnitude from the right to the left hemisphere, the specificity of the IHI manipulation was not proved yet. We presented additional results related to changes in corticospinal excitability as assessed by MEPs in the left FDI in response to the CS over the right hemisphere (Figure 4—figure supplements 3B). A one-way rmANOVA for sessions (five levels: REST, NoFB, HIGH, MID, and LOW) revealed no significant differences in the left MEP amplitude between all sessions (F_(4,101)_ = 0.39, *p* = 0.813, η^2^ = 0.02). Furthermore, in order to investigate whether changes in IHI from the right to the left hemisphere and changes in corticospinal excitability in the right hemisphere are independent phenomena, a direct statistical comparison between the normalized IHI and the normalized left MEP was applied. During the triggered TMS trials, a two-way rmANOVA for sessions (three levels: HIGH, MID, and LOW) and indices (two levels: increase in IHI and left MEP amplitude) revealed a significant main effect for sessions (F_(2, 122)_ = 3.96, *p* = 0.022, η^2^ = 0.06), but no main effect for indices (F_(1, 122)_ = 1.11, *p* = 0.294, η^2^ = 0.01) and for interaction (F_(2, 122)_ = 1.65, *p* = 0.195, η^2^ = 0.03). A post-hoc two-tailed paired t-test in the increase in IHI showed significant differences between HIGH and MID sessions (Cohen's *d* = 1.06, *p* = 0.011), and between HIGH and LOW sessions (Cohen's *d* = 1.42, *p* < 0.001). In contrast, a post-hoc two-tailed paired t-test in the left MEP amplitude showed no significant differences between HIGH and LOW sessions (Cohen's *d* = 0.16, *p* = 1.00). During the nontriggered TMS trials, a two-way rmANOVA for sessions (three levels: HIGH, MID, and LOW) and indices (two levels: increase in IHI and eft MEP amplitude) revealed no main effect for sessions (F_(2, 122)_ = 0.95, *p* = 0.390, η^2^ = 0.02), for indices (F_(1, 122)_ = 0.81, *p* = 0.369, η^2^ = 0.01) and for interaction (F_(2, 122)_ = 0.49, *p* = 0.614, η^2^ = 0.01). Therefore, we successfully revealed that modulating the excitability of the right hemisphere changes IHI from the right to the left hemisphere, but a significant interaction between IHI modulation and changes in the left MEP amplitude was not observed.”

[Lines 1276-1317] “Our study has several limitations that need to be considered when interpreting the results. … Additionally, we successfully revealed that modulating the excitability of the right hemisphere changes IHI from the right to the left hemisphere, but the specificity of the IHI manipulation could not be fully demonstrated (Figure 4—figure supplements 3B and C). … To confirm whether changes in interhemispheric inhibition and corticospinal excitability are independent phenomena, the CS effect should have been kept constant across the conditions by adjusting the CS intensity to evoke left MEP of 1 mV peak-to-peak amplitude. Because such an experiment was not possible in the present study due to time limitation (the experimental time would need to be doubled to 5–6 h to perform such an experiment), future studies will be needed.”

[Lines 1853-1860] “The statistical differences between corticospinal excitability as assessed by MEPs in the left FDI in response to the CS over the right hemisphere was tested by a one-way rmANOVA for sessions (five levels: REST, NoFB, HIGH, MID, and LOW). To investigate whether changes in interhemispheric inhibition and corticospinal excitability are independent phenomena, a direct statistical comparison using a two-way rmANOVA for sessions (three levels: HIGH, MID, and LOW) and indices (two levels: increase in IHI and eft MEP amplitude) was applied in the triggered and non-triggered TMS trials.”

4) Bidirectional control of IHI would need to be significantly demonstrated above and below the resting baseline. Although the study shows bidirectional control of SMR power (in the ipsilateral hemisphere), this is strictly not the case for IHI (which shows only significant increases relative to rest).

Although we successfully demonstrated the up- and down-regulate the ipsilateral SMR-ERD (i.e., bidirectional changes) at EEG-level, the bidirectional changes in IHI from the right to the left hemisphere were not statistically shown since there is no significant difference in MID and LOW sessions (Figure 4B). Therefore, we have revised the expression of the “bidirectional” modulation in IHI and added the future experiment in the Limitation.

[Lines 1277-1282] “Although we successfully demonstrated the bidirectional changes in the ipsilateral SMR-ERD, the bidirectional changes in IHI from the right to the left hemisphere were not statistically significant since there was no significant difference between the MID and LOW sessions (Figure 4B). The current results should prompt future studies to ascertain whether down-regulation of IHI (i.e., disinhibition) during MI and up-regulation of IHI at rest in the non-dominant hemisphere (intact hemisphere) are possible in a similar setup.”

[Lines 1350-1355] “In conclusion, we presented an innovative approach to manipulate the state of IHI, by directly and bidirectionally modulating SMR-ERDs in a spatially bivariate BCI-based neurofeedback paradigm.”

5) The result section is difficult to understand because context information is often missing since the detailed Methods are reported at the end. Please revise the results to improve accessibility for the reader.

According to this suggestion, we have revised throughout the manuscript and swapped the order in the results to improve accessibility for the reader.

e.g., explanations of “triggered trials” and “waiting time” [Lines 247-251], stimulus intensities during TMS experiment [Lines 351-356], explanations of each five sessions including REST and NoFB sessions [Lines 218-220], and how to evaluate interhemispheric functional connectivity [Lines 804-808].

In addition, values corresponding to each statistical figure are presented in each table (Supplementary File 1) for readability.

6) Some statements like "suggesting that the volitional control of SMR-ERDs in the closed-loop environment differs from spontaneous SMR fluctuations" require a direct statistical comparison

We have revised to the correct sentences as below.

In addition, we performed a direct statistical comparison and replaced the result in Figure 4—figure supplements 2.

[Lines 455-554] “A two-way rmANOVA for sessions (three levels: HIGH, MID, and LOW) and trials (two levels: triggered TMS trials and non-triggered TMS trials) revealed a significant main effect for sessions (F_(2, 117)_ = 5.08, *p* = 0.008, η^2^ = 0.08), but no main effect for trials (F_(1, 117)_ = 0.32, *p* = 0.575, η^2^ < 0.01) and for interaction (F_(2, 117)_ = 1.82, *p* = 0.167, η^2^ = 0.03). Thus, no significant difference in IHI between the volitional control of SMR-ERDs in the closed-loop environment differs from spontaneous SMR fluctuations was observed. A post-hoc two-tailed paired t-test showed significant differences between HIGH and MID sessions (Cohen's *d* = 0.94, *p* = 0.017), and between HIGH and LOW sessions (Cohen's *d* = 1.36, *p* < 0.001) in the triggered TMS trials. In contrast, in the non-triggered TMS trials, a posthoc two-tailed paired t-test showed no significant differences between HIGH and MID sessions (Cohen's *d* = 0.20, *p* = 1.00), and between HIGH and LOW sessions (Cohen's *d* = 0.28, *p* = 1.00).”

[Lines 1850-1853] “To investigate the influence of spontaneous SMR fluctuations on IHI, a two-way rmANOVA for sessions (HIGH, MID, and LOW) and trials (triggered TMS trial and non-triggered TMS trial) as the withinparticipant factors.”

7) Revise correlation analyses to properly account for within versus between-subject variability

In accordance with your comments regarding the across-participant corrections between IHI magnitude and bilateral EEG patterns, we performed a repeated measures correlation since it was not reasonable to combine different sessions of different participants. Simple correlation is often applied to non-independent observations or aggregated data. A previous study argued that it may produce biased, specious results due to violation of independence and/or differing patterns (across-participants vs. within-participants) (Bakdash and Marusich, 2017).

We also calculated within-participant corrections between IHI magnitude and bilateral EEG patterns in accordance with the suggestion by Reviewer 3. We have added the overall significance of the 22 individual correlations using a chi-squared test and the distribution of correlation coefficients across all participants. In the analysis, data from HIGH, MID, and LOW sessions were aggregated as independent observations. We have integrated the above suggestions and modified the text as follows.

[Lines 1772-1776] “In the across-participant corrections between contralateral or ipsilateral SMR-ERDs and IHI magnitude, we performed a repeated measures correlation. A previous study argued that it may produce biased, specious results due to violation of independence and/or differing patterns across-participants versus withinparticipants (Bakdash and Marusich, 2017). “

[Lines 717-803] “In the across-participant correlation between IHI magnitude and contralateral or ipsilateral SMR-ERDs, we performed a repeated measures correlation analysis (Bakdash and Marusich, 2017). We found a significant correlation in the ipsilateral SMR-ERD (r_rm_ = 0.436, *p* = 0.004; Figure 5B), but not in the contralateral SMR-ERD (r_rm_ = 0.008, *p* = 0.960; Figure 5C).”

8) It remains an open question, whether the more sophisticated neurofeedback approach (of modulating both hemispheres separately) was indeed necessary to achieve the findings of this work.

Spatially bivariate BCI-based neurofeedback enables us to modulate the ipsilateral SMR-ERD bidirectionally while maintaining constant contralateral SMR-ERD. It provided new findings that the ipsilateral SMR-ERD influenced the IHI magnitude. As you mentioned, it remains an open question, whether the conventional BCI-based neurofeedback that exploits SMR-ERD from one hemisphere is substitutable to achieve the findings of this work. In this context, in case the contralateral SMR-ERD was up- or down-regulated instead of maintained at a moderate level, it is unclear whether the ipsilateral SMR-ERD was modulated bidirectionally and whether the IHI changed with it. The study performed a total of 11 sessions, including REST and NoFB, and the combinations of SMR-ERDs in both hemispheres (3 conditions vs. 3 conditions) may be worth evaluating in the future.

We have added future directions as follows:

[Lines 1329-1339] “Moreover, in case the contralateral SMR-ERD was up- or down-regulated instead of maintained at a moderate level, it is unclear whether the ipsilateral SMR-ERD was modulated bidirectionally and whether the IHI changed with it. The study performed a total of 11 sessions, including REST and NoFB, and the combinations of SMR-ERDs in both hemispheres (3 conditions vs. 3 conditions) may be worth evaluating in the future. In addition, it is not clear whether the bivariate neurofeedback approach indeed contributed to the observed findings since there is no comparison to the conventional BCI-based neurofeedback that exploits SMR-ERD from one hemisphere. A future study is required to investigate whether the conventional univariate neurofeedback that exploits SMR-ERD from one hemisphere achieves bidirectional modulation of the ipsilateral SMR-ERD and accompanying IHI magnitude.”

9) The dynamic range of SMR modulation occurs mainly in the ipsilateral hemisphere (in terms of bidirectional control). The contralateral hemisphere consistently shows increased SMR power relative to rest, with no bidirectional modulation. Please explain why this was the case and discuss potential implications

We aimed to modulate the ipsilateral hemisphere bidirectionally while maintaining constant the contralateral excitability using our spatially bivariate BCI-based neurofeedback. To evaluate the effect of SMR modulation in the ipsilateral hemisphere, participants always aimed to maintain constant contralateral SMR-ERD. As written in the manuscript ("Spatially bivariate BCI-based neurofeedback"), visual feedback was provided on a computer screen in the form of cursor movements in a two-dimensional coordinate, in which x- and y-axis corresponded to the degree of the ipsilateral and contralateral SMR-ERD, respectively. For example, in HIGH session, participants were instructed to move the cursor toward the middle right (x > 0, y = 0) in the two-dimensional coordinate, the position showed a strong SMR-ERD in the ipsilateral hemisphere and moderate SMR-ERD in the contralateral hemisphere. On the contrary, in LOW session, participants were instructed to move the cursor toward the middle left (x < 0, y = 0) in the two-dimensional coordinate, the position showed a low SMR-ERD (or strong SMR-ERS) in the ipsilateral hemisphere and moderate SMR-ERD in the contralateral hemisphere. Therefore, spatially bivariate BCI-based neurofeedback enables us to modulate the ipsilateral SMR-ERD bidirectionally while maintaining constant contralateral SMR-ERD in the three sessions (i.e., HIGH, MID and LOW). In fact, Figures 2A and B showed the intended results. To help readers’ understanding, we prepared a sample video as Video 1.

The above is the answer to your point, but we notice that we need to describe the experimental limitation regarding (1) handedness and (2) the other patterns of the contralateral excitability. In this study, all participants were right-handed (Laterality Quotient: 72.2 ± 30.9%), as assessed by the Edinburgh Inventory (Oldfield, 1971), to eliminate bias due to the dominance. If a left-handed participant performs the same experiment, the modulation effect via BCI-based neurofeedback may be different. Moreover, in case the contralateral SMR-ERD was up- or downregulated instead of maintained at a moderate level, it is unclear whether the ipsilateral SMR-ERD was modulated bidirectionally and whether the IHI changed with it. The study performed a total of 11 sessions, including REST and NoFB, and the combinations of SMR-ERDs in both hemispheres (3 conditions vs. 3 conditions) may be worth evaluating in the future.

[Line 1325-1334] “For the limitations of the volitional control of SMR-ERD during unilateral MI, all participants were right-handed (Laterality Quotient: 72.2 ± 30.9%), as assessed by the Edinburgh Inventory (Oldfield, 1971), to eliminate bias due to the dominance. If a left-handed participant performs the same experiment, the modulation effect via BCI-based neurofeedback may be different. Moreover, in case the contralateral SMR-ERD was up- or downregulated instead of maintained at a moderate level, it is unclear whether the ipsilateral SMR-ERD was modulated bidirectionally and whether the IHI changed with it. The study performed a total of 11 sessions, including REST and NoFB, and the combinations of SMR-ERDs in both hemispheres (3 conditions vs. 3 conditions) may be worth evaluating in the future.”

10) Why was the gender ratio of the subjects (1:11, F:M) so highly skewed? This should be explained, and potential limitations should be discussed

The study participants were freely recruited from the university without any criteria for sex; there was no experimental plan to equalize the sex ratio. Because there were few female participants in the study, it is unclear whether our findings can be generalized to females. Previous studies revealed that the brain microstructure, including corpus callosum, differed between men and women (Dunst et al., 2014; Menzler et al., 2011). It is unknown whether these sex differences influenced our IHI study and additional studies are required to determine this.

We added this limitation as follows:

[Lines 1341-1347] “Finally, the sex ratio in this study was the consequence of recruiting freely from the university; there was no experimental plan to equalize the sex ratio. Because there were few female participants in the study, it is unclear whether our findings can be generalized to females. Previous studies revealed that the brain microstructure, including corpus callosum, differed between men and women (Dunst et al., 2014; Menzler et al., 2011). It is unknown whether these sex differences influenced our IHI study and additional studies are required to determine this.”

11) The discussion of the neurophysiological mechanisms mediating IHI would benefit from more depth and a better embedding into the literature.

We have added the information related to the neurophysiological mechanisms mediating IHI. In addition, the review by Carson provided new insight into reassessing the inter-hemispheric competition model. The alternative conceptual framework, which states that IHI acts to sculpt the output of neural circuits by co-opting the two cerebral hemispheres, is shown to be consistent with the known associations between the structural integrity of callosal projections and the magnitude of the motor deficits after stroke. This may emphasize the benefit of our neurofeedback approach in the context of neurorehabilitation.

With have added several additional references and revised the discussion as follows:

[Lines 1150-1155] “This may indicate that IHI and Network-intensity simply reflect the activity of different neural populations; interhemispheric projections act via surround/lateral inhibition in the sensory and motor cortices (Carson, 2020), while the synchrony at the EEG level reflects changes in the activity of local interactions between pyramidal neurons and interneurons in the thalamocortical loops (Pfurtscheller and Lopes da Silva, 1999).”

[Lines 1265-1270] “Furthermore, a previous review advocated the conceptual framework and suggested that a fundamental role of IHI is to support the narrowing of excitatory focus by co-opting the capacities of the two cerebral hemispheres (Carson, 2020). This framework is consistent with the known associations between the structural integrity of callosal projections and the magnitude of the motor deficits after stroke (Auriat et al., 2015; Granziera et al., 2012; Koh et al., 2018).”

[Lines 1307-1311] “Previous studies suggested that the IHI magnitude and the corticomotor excitability were modulated independently (Morishita et al., 2011, 2012). Although it is difficult to directly compare these results due to the significantly different motor tasks and experimental settings, the assertion that the projections from the transcallosal and the corticospinal pathways are distinct supports our findings.”

Reviewer #1 (Recommendations for the authors):The manuscript reports the interesting phenomenon that selectively modulating the excitability of the right hemisphere changes interhemispheric inhibition from the right to the left hemisphere. While this is interesting, I'm wondering how the interhemispheric result relates to changes in corticospinal excitability as assessed via MEPs in response to the conditioning stimulus.I understood that the CS has been chosen such that TS MEPs are reduced by approximately 50%. Was that done while participants were at rest or while they performed the MI task /BCI control?If the CS intensity was determined at rest, did the authors analyse the CS MEPs and particularly how they change from IHI at rest to IHI during the MI/BCI control? Does IHI change independently of MEP modulation in response to the CS? This would be an important analysis also for the shuffled non-triggered trials since it might reveal whether corticospinal excitability and IHI can be controlled independently.Please note that there has been some debate in the field whether or not the CS effect has to be kept constant across conditions. Thus, if there are differences in CS MEP amplitudes between the sessions, the authors might want to address this methodological issue in the discussion.

We appreciate your valuable comments. We have added additional results related to changes in corticospinal excitability as assessed by MEPs in the left FDI in response to the conditioning stimulus over the right hemisphere (Figure 4—figure supplements 3B). A direct statistical comparison between increase in IHI and left MEP amplitude is shown in Figure 4—figure supplements 3C. To compare the two indices, the percent changes based on the values in NoFB session were calculated. The stimulus intensity of the CS during the experiment was set such that the CS evoked MEP of peak-to-peak amplitude from the left FDIs (SI_1mV_) “at rest” (Lines 1482-1484, “The stimulus intensities of the left and right M1 to evoke MEP …”). In addition, we analyzed the non-triggered trials to show the CS effect (Figure 4—figure supplements 3D).

Based on the additional results, the MEP amplitudes in the left FDI elicited by CS did not vary across sessions, indicating that the modulation of the ipsilateral SMR-ERD caused no changes in corticospinal excitability in the contralateral side. Although we successfully revealed that modulating the excitability of the right hemisphere changes IHI from the right to the left hemisphere, a significant interaction between IHI modulation and changes in the left MEP amplitude was not observed. Thus, the specificity of the IHI manipulation could not be fully demonstrated. To confirm whether the participants learned pathway-specific IHI manipulation, the CS effect should have been kept constant across the conditions by adjusting the CS intensity to evoke left MEP of 1 mV peak-to-peak amplitude. Because such an experiment was not possible in the present study due to the time limitation (the experimental time would need to be doubled to 5–6 h to perform such an experiment), further studies are warranted.

We have toned down our results and conclusions, such as the use of the term "pathway-specific" manipulation, since our experiment did not directly demonstrate whether changes in interhemispheric inhibition and corticospinal excitability are independent phenomena. We have revised the manuscript to include these additional analyses, figures, and limitations.

[Lines 555-599] “Although we revealed that modulating the excitability of the right hemisphere changes IHI magnitude from the right to the left hemisphere, the specificity of the IHI manipulation was not proved yet. We presented additional results related to changes in corticospinal excitability as assessed by MEPs in the left FDI in response to the CS over the right hemisphere (Figure 4—figure supplements 3B). A one-way rmANOVA for sessions (five levels: REST, NoFB, HIGH, MID, and LOW) revealed no significant differences in the left MEP amplitude between all sessions (F_(4,101)_ = 0.39, *p* = 0.813, η^2^ = 0.02). Furthermore, in order to investigate whether changes in IHI from the right to the left hemisphere and changes in corticospinal excitability in the right hemisphere are independent phenomena, a direct statistical comparison between the normalized IHI and the normalized left MEP was applied. During the triggered TMS trials, a two-way rmANOVA for sessions (three levels: HIGH, MID, and LOW) and indices (two levels: increase in IHI and left MEP amplitude) revealed a significant main effect for sessions (F_(2, 122)_ = 3.96, *p* = 0.022, η^2^ = 0.06), but no main effect for indices (F_(1, 122)_ = 1.11, *p* = 0.294, η^2^ = 0.01) and for interaction (F_(2, 122)_ = 1.65, *p* = 0.195, η^2^ = 0.03). A post-hoc two-tailed paired t-test in the increase in IHI showed significant differences between HIGH and MID sessions (Cohen's *d* = 1.06, *p* = 0.011), and between HIGH and LOW sessions (Cohen's *d* = 1.42, *p* < 0.001). In contrast, a post-hoc two-tailed paired t-test in the left MEP amplitude showed no significant differences between HIGH and LOW sessions (Cohen's *d* = 0.16, *p* = 1.00). During the nontriggered TMS trials, a two-way rmANOVA for sessions (three levels: HIGH, MID, and LOW) and indices (two levels: increase in IHI and eft MEP amplitude) revealed no main effect for sessions (F_(2, 122)_ = 0.95, *p* = 0.390, η^2^ = 0.02), for indices (F_(1, 122)_ = 0.81, *p* = 0.369, η^2^ = 0.01) and for interaction (F_(2, 122)_ = 0.49, *p* = 0.614, η^2^ = 0.01).”

Therefore, we successfully revealed that modulating the excitability of the right hemisphere changes IHI from the right to the left hemisphere, but a significant interaction between IHI modulation and changes in the left MEP amplitude was not observed.

[Lines 1276-1317] “Our study has several limitations that need to be considered when interpreting the results. … Additionally, we successfully revealed that modulating the excitability of the right hemisphere changes IHI from the right to the left hemisphere, but the specificity of the IHI manipulation could not be fully demonstrated (Figure 4—figure supplements 3B and C). … To confirm whether changes in interhemispheric inhibition and corticospinal excitability are independent phenomena, the CS effect should have been kept constant across the conditions by adjusting the CS intensity to evoke left MEP of 1 mV peak-to-peak amplitude. Because such an experiment was not possible in the present study due to time limitation (the experimental time would need to be doubled to 5–6 h to perform such an experiment), future studies will be needed.”

[Lines 1853-1860] “The statistical differences between corticospinal excitability as assessed by MEPs in the left FDI in response to the CS over the right hemisphere was tested by a one-way rmANOVA for sessions (five levels: REST, NoFB, HIGH, MID, and LOW). To investigate whether changes in interhemispheric inhibition and corticospinal excitability are independent phenomena, a direct statistical comparison using a two-way rmANOVA for sessions (three levels: HIGH, MID, and LOW) and indices (two levels: increase in IHI and eft MEP amplitude) was applied in the triggered and non-triggered TMS trials.”

The result section is difficult to understand because context information is often missing since the detailed Methods are reported at the end. Please revise the results to improve accessibility for the reader. In that regard, I would recommend swapping the order in the result section and presenting first the EEG states and then the associated IHI measures.

According to your suggestion, we have revised throughout the manuscript and swapped the order in the results to improve accessibility for the reader. e.g., explanations of “triggered trials” and “waiting time” [Lines 247-251], stimulus intensities during TMS experiment [Lines 351-356], explanations of each five sessions including REST and NoFB sessions [Lines 218-220], and how to evaluate interhemispheric functional connectivity [Lines 804-808].

In addition, values corresponding to each statistical figure are presented in each table (Supplementary File 1) for readability.

What is the rationale for using right finger MI as the no-FB condition? Please explain

The NoFB session served as a baseline control to examine changes in IHI magnitude during MI without any influence of visual input and volitional control. In the three neurofeedback HIGH, MID, and LOW sessions, participants received visual feedback during right finger MI and learned to modulate the sensorimotor activity patterns using their own volition. In other words, these three sessions are influenced by visual input and volitional control, in addition to performing MI (Takemi et al., 2018).

We added explanation of the rationale of NoFB session in the Results section as follows:

[Lines 215-220] “In the three neurofeedback sessions (i.e., HIGH, MID, and LOW), participants received visual feedback during right finger MI and tried to modulate the sensorimotor activity patterns. The NoFB session served as a baseline to examine the IHI magnitude without visual stimuli and volitional control. The REST session also served as a control to determine the intrinsic IHI magnitude at rest.”

Line 257ff: there seems to be a mistake: «one-way rmANOVA for sessions (three levels: HIGH, MID, and LOW) revealedsignificant differences F(2,44) = 1.43, p < 0.248,…» should probably read « revealed no significant differences».

Corrected. [Lines 451-554]

Line 260 ff there seems to be a mistake: Although the IHI magnitude in the LOW session was larger than that in the HIGH and LOW sessions.… replace second LOW by MID.I liked the control analysis of the authors shown in suppl. Figure 3-2, However, the conclusion that the LOW,MID,HIGH data shown in Figure 3B is different from the data shown in suppl. Figure 3-2 requires a direct statistical comparison.

We have revised to the correct sentences as below.

In addition, we performed a direct statistical comparison and replaced the result in Figure 4—figure supplements 2

[Lines 455-554] “A two-way rmANOVA for sessions (three levels: HIGH, MID, and LOW) and trials (two levels: triggered TMS trials and non-triggered TMS trials) revealed a significant main effect for sessions (F_(2, 117)_ = 5.08, *p* = 0.008, η^2^ = 0.08), but no main effect for trials (F_(1, 117)_ = 0.32, *p* = 0.575, η^2^ < 0.01) and for interaction (F_(2, 117)_ = 1.82, *p* = 0.167, η^2^ = 0.03). Thus, no significant difference in IHI between the volitional control of SMR-ERDs in the closed-loop environment differs from spontaneous SMR fluctuations was observed. A post-hoc two-tailed paired t-test showed significant differences between HIGH and MID sessions (Cohen's *d* = 0.94, *p* = 0.017), and between HIGH and LOW sessions (Cohen's *d* = 1.36, *p* < 0.001) in the triggered TMS trials. In contrast, in the non-triggered TMS trials, a posthoc two-tailed paired t-test showed no significant differences between HIGH and MID sessions (Cohen's *d* = 0.20, *p* = 1.00), and between HIGH and LOW sessions (Cohen's *d* = 0.28, *p* = 1.00).”

[Lines 1850-1853] “To investigate the influence of spontaneous SMR fluctuations on IHI, a two-way rmANOVA for sessions (HIGH, MID, and LOW) and trials (triggered TMS trial and non-triggered TMS trial) as the withinparticipant factors.”

Figure 4C: I'm not convinced that presenting the laterality index adds any information to Figure 4 A and B. In fact, I found the caption confusing since it seems to suggest that (i) the negative LI for LOW results from contralateral activity which is inconsistent with the topoplot that shows high activity in the ipsilateral hemisphere; and (ii) the positive LI for HIGH results from ipsilateral activity which is also inconsistent with the topo which shows desynchronisation in the ipsilateral hemisphere. Given the shown in panels A and B, the LI measure seems inappropriate since there are virtually no activity changes in the contralateral hemisphere while the ipsilateral hemisphere shows either synchronized or desynchronized SMR activity.

We agree that the laterality index (LI) does not provide additional important information to Figures 2A and B. To avoid confusion for the readers, the results and descriptions related to the LI have been deleted from the manuscript.

Associations between IHI magnitude and bilateral EEG patterns: The authors might want to consider calculating a repeated measures correlation because it does not seem meaningful to pool data across the LOW, MID, HIGH sessions.

In accordance with your comments regarding the across-participant corrections between IHI magnitude and bilateral EEG patterns, we performed a repeated measures correlation since it was not reasonable to combine different sessions of different participants. Simple correlation is often applied to non-independent observations or aggregated data. A previous study argued that it may produce biased, specious results due to violation of independence and/or differing patterns (across-participants vs. within-participants) (Bakdash and Marusich, 2017).

We also calculated within-participant corrections between IHI magnitude and bilateral EEG patterns in accordance with the suggestion by Reviewer 3. We have added the overall significance of the 22 individual correlations using a chi-squared test and the distribution of correlation coefficients across all participants. In the analysis, data from HIGH, MID, and LOW sessions were aggregated as independent observations. We have integrated the above suggestions and modified the text as follows.

[Lines 1772-1776] “In the across-participant corrections between contralateral or ipsilateral SMR-ERDs and IHI magnitude, we performed a repeated measures correlation. A previous study argued that it may produce biased, specious results due to violation of independence and/or differing patterns (across-participants vs. withinparticipants) (Bakdash and Marusich, 2017).”

[Lines 717-803] “In the across-participant correlation between IHI magnitude and contralateral or ipsilateral SMR-ERDs, we performed a repeated measures correlation analysis (Bakdash and Marusich, 2017). We found a significant correlation in the ipsilateral SMR-ERD (r_rm_ = 0.436, *p* = 0.004; Figure 5B), but not in the contralateral SMR-ERD (r_rm_ = 0.008, *p* = 0.960; Figure 5C)”

DiscussionLine 470ff: The first sentence of the discussion follows a reversed logic. Participants did not consciously self-regulate IHI but they modulated their EEG pattern based on the closed-loop spatially bivariate BCI-based neurofeedback. Changes in IHI were the consequence of self-regulating the EEG activity pattern. Please revise throughout the discussion.

We have revised the explanations throughout the Discussion. [Lines 1024-1046 (“In the present study, we aimed…”), Lines 1350-1355 (“In conclusion, we presented an innovative approach to…”)]. Similarly, the claim was simply used “manipulate” or “manipulation” instead of “consciously/voluntary self-manipulate” or “consciously/voluntary selfmanipulation”.

The discussion of the neurophysiological mechanisms mediating IHI could be a bit more in-depth. The review of Carson RG. J Physiol. 2020 Nov;598(21):4781-4802 might help.

We appreciate for suggesting the study by Carson. We have added the information related to the neurophysiological mechanisms mediating IHI. In addition, the review by Carson provided new insight into reassessing the interhemispheric competition model. The alternative conceptual framework, which states that IHI acts to sculpt the output of neural circuits by co-opting the two cerebral hemispheres, is shown to be consistent with the known associations between the structural integrity of callosal projections and the magnitude of the motor deficits after stroke. This may emphasize the benefit of our neurofeedback approach in the context of neurorehabilitation.

With have added several additional references and revised the discussion as follows:

[Lines 1150-1155] “This may indicate that IHI and Network-intensity simply reflect the activity of different neural populations; interhemispheric projections act via surround/lateral inhibition in the sensory and motor cortices (Carson, 2020), while the synchrony at the EEG level reflects changes in the activity of local interactions between pyramidal neurons and interneurons in the thalamocortical loops (Pfurtscheller and Lopes da Silva, 1999).”

[Lines 1265-1270] “Furthermore, a previous review advocated the conceptual framework and suggested that a fundamental role of IHI is to support the narrowing of excitatory focus by co-opting the capacities of the two cerebral hemispheres (Carson, 2020). This framework is consistent with the known associations between the structural integrity of callosal projections and the magnitude of the motor deficits after stroke (Auriat et al., 2015; Granziera et al., 2012; Koh et al., 2018).”

[Lines 1307-1311] “Previous studies suggested that the IHI magnitude and the corticomotor excitability were modulated independently (Morishita et al., 2011, 2012). Although it is difficult to directly compare these results due to the significantly different motor tasks and experimental settings, the assertion that the projections from the transcallosal and the corticospinal pathways are distinct supports our findings.”

The authors argue that "This multimodal EEG-TMS approach proves that the bilateral α and β activities observed in the current study served to regulate the inhibition and facilitation of inhibitory functional coupling of interhemispheric interaction over motor cortices.» This argument would be strengthened if the interhemispheric effects were independent of the general excitability of the ipsilateral hemisphere as reflected in CS MEP amplitudes. Only then, the authors can argue that their intervention specifically modulated interhemispheric circuits.

As stated in to our response to the first comment, our results were not adequate to confirm that the changes in IHI and changes in the corticospinal excitability were independent of each other. Therefore, we have deleted the sentence. In addition, the methodological limitations of the study have been revised, as stated in our response to the first comment.

Lines 538-544: Explanatory theoretical model: I cannot follow the argument of the authors. Is it possible that IHI and "network intensity" simply reflect the activity of different neural populations? IHI effects are rather specific and I doubt that this could be picked up via general EEG markers.

We agree with your concern that different neural populations might be involved in IHI and Network-intensity. In addition, Network-intensity reflects the bidirectional synchrony between the two hemispheres, while IHI reflects oneway (right-to-left) inhibitory effects. Therefore, we have deleted the sentences to simplify the Discussion section.

[Lines 1142-1157] “However, even fMRI-based neurofeedback with high spatial resolution, based on blood oxygenation level-dependent signal, cannot easily distinguish between the excitatory and the inhibitory activities from each region (Moon et al., 2021). Additionally, neurofeedback using interhemispheric functional connectivity has the possibility to counteract the modulation of IHIs from the right to left hemisphere and from the left to right hemisphere. Our results showed no significant difference in Network-intensity during MI between HIGH and LOW sessions (Figure 5D), indicating that interhemispheric functional connectivity at the EEG level was not changed, whereas IHI was modulated. This may indicate that IHI and Network-intensity simply reflect the activity of different neural populations; interhemispheric projections act via surround/lateral inhibition in the sensory and motor cortices (Carson, 2020), while the synchrony at the EEG level reflects changes in the activity of local interactions between pyramidal neurons and interneurons in the thalamocortical loops (Pfurtscheller and Lopes da Silva, 1999). Furthermore, it is possible that Network-intensity reflects the degree of bidirectional synchrony between the two motor cortices, while IHI in this study reflects one-way (right-to-left) inhibitory effects.”

[Lines 1157-1159] “Therefore, it is unclear whether fMRI-based neurofeedback using interhemispheric functional connectivity is linked to ipsilateral excitability, and whether it modulates IHI magnitude.”

Line 561ff This seems to be consistent «Furthermore, the effect size of IHI manipulation in the current study (Cohen's d = 1.50) was approximately 1-2-fold higher than that of representative tDCS (Cohen's d = 1.55)…».

We have revised this sentence as follows:

[Lines 1168-1171] “Furthermore, the effect size of IHI manipulation in the current study (Cohen's *d* = 1.50) was comparable with that of representative tDCS (Cohen's *d* = 1.55) (Williams et al., 2010) and superior to that of rTMS studies (Gilio et al., 2003) (Cohen's *d* = 0.80; note that it is a read from the graph).”

Discussion of IHI in stroke: As stated by the authors, it is somewhat controversial whether IHI is a useful therapy target early after stroke. However, their EEG-neurofeedback approach allows to selectively up- or down-regulate one hemisphere which might be highly relevant in the context of stroke recovery both from a basic science and application perspective. It might be worthwhile to expand this aspect of the discussion.

We thank your encouragement. Following the original sentences, we complemented the pointed aspects.

[Lines 1270-1272] “Thus, our spatially bivariate EEG-neurofeedback approach might be highly relevant to application perspective in the context of stroke recovery as well as a basic science perspective.”

Reviewer #2 (Recommendations for the authors):Although the Introduction is comprehensive in terms of the mechanistic relation between EEG and TMS variables, it does not sufficiently include the "clinical" significance of modifying IHI e.g. in terms of rehabilitation potential. It would be nice to include a few references in clinical populations where IHI and/or SMR are impacted, or where IHI/SMR have been used as prognostic markers of motor recovery. I would also include a sentence or two in the Abstract underlining the potential clinical significance of manipulating IHI.

During the initial review of this manuscript, we received the following comment from a senior editor: “the introduction focused on the relevance of the work for stroke rehabilitation seemed a stretch and, in some sense, to weaken the work given that you do not include any behavioral outcome measures, and that the rebalancing argument has proven somewhat contentious”.

Because Reviewer 1 also provided a similar suggestion, we added the clinical significance of our results. Thus, we addressed the rehabilitation issue in the Discussion section and focused on the core question of our work and relevant literature in the Introduction section.

We have revised the manuscript to state the importance of our findings in the context of neurorehabilitation as follows:

[Lines 1252-1260] “In stroke patients, the interhemispheric imbalance/competition model predicts the presence of asymmetry in the interhemispheric sensorimotor network, with excessive inhibition from the non-affected hemisphere limiting motor performance and maximal recovery (Duque et al., 2005; Murase et al., 2004). Therefore, guiding effective inhibitory interhemispheric network represented by IHI to the appropriate pattern through both targeted up-conditioning in the affected hemisphere and down-conditioning in the non-affected hemisphere may contribute to reduced abnormal IHI and enhanced achievable functional recovery (Chieffo et al., 2013; Di Pino et al., 2014; Dong et al., 2006; Hummel and Cohen, 2006).”

In terms of related studies combining the TMS/EEG/BCI trio, the work by Ros et al. 2010 was one of the first to demonstrate that MEP measures of cortical excitability and inhibition (MEP, SICI) can be modulated using closed-loop feedback of oscillations from the motor cortex.Ros, T., Munneke, M.A.M., Ruge, D., Gruzelier, J.H. and Rothwell, J.C. (2010), Endogenous control of waking brain rhythms induces neuroplasticity in humans. European Journal of Neuroscience, 31: 770-778. https://doi.org/10.1111/j.1460-9568.2010.07100.x

Thank you for introducing the previous study. We cited the recommended literature.

[Lines 1056-1073] “Using spatially bivariate BCI-based neurofeedback enables participants to volitionally increase or decrease (bidirectional) the ipsilateral sensorimotor excitability, while maintaining constant contralateral sensorimotor excitability. Ipsilateral SMR-ERD (but not contralateral), reflecting IHI magnitude (Figure 2B) and their significant correlation (Figure 5C), were compatible with previous studies (Haegens et al., 2011; Madsen et al., 2019; Ros et al., 2010; Sauseng et al., 2009; Takemi et al., 2013; Thies et al., 2018; Zarkowski et al., 2006). The … Furthermore, another EEG-TMS study using neurofeedback demonstrated that the endogenous suppression of α rhythms at rest can produce robust increase in MEP amplitude and decrease in short-interval intracortical inhibition (Ros et al., 2010).”

It is not clear what behavioural and neural variable the authors are referring to in the first Results section starting with. "We confirmed the difficulties of each session" (line 171).

We have added the explanations in the Results (the same thing was pointed out by Reviewer 1).

[Lines 247-251] “To quantify the difficulties in each neurofeedback session, two indices were examined: (1) the number of triggered trials represents the BCI performance (i.e., success trials) since TMS was only triggered when the EEG signal exceeded the predetermined SMR-ERD threshold; (2) mean waiting time for a triggered event from task onset, which yields a range of 0–5 s (triggered TMS timing).”

Line 171: "The mean value of the sum of the triggered trials". Mean and sum have different definitions, do you mean 'the mean value of all the triggered trials'?

We have corrected the relevant part.

[Lines 251-255] “The mean values of all triggered trials (± 1 SD) of the neurofeedback sessions were 14.2 ± 4.7 trials and post-hoc paired t-tests following a one-way repeated-measures ANOVA (rmANOVA) revealed no significant difference between the three sessions (all *p* > 0.05; HIGH: 12.0 ± 5.0 trials, MID: 16.1 ± 3.8 trials, LOW: 15.9 ± 5.4 trials).”

There is no reference to what the abbreviation 'TS' stands for (I am assuming it is "test stimulus")

We have already written the abbreviation in Line 342. We also added it in the “abbreviation” section. [Line 58]

Line 219: The manipulation range of IHI (i.e., the difference between HIGH and LOW sessions) was 32.6. However elsewhere the means (Line 222/ 223) are separately indicated as HIGH: 62.0 and LOW: 96.9. Their difference (96.9 – 62.0) is 34.9.

Due to a few corrupted data in HIGH session (Lines 1401-1403), the manipulation range was not calculated in two participants, so the two values were different. The MEP amplitude in LOW session excluding two participants was 94.6%. Thus, the difference between HIGH and LOW sessions was 32.6% which was consistent with written information. To avoid the confusion of readers, we have corrected the sentence as follows:

[Lines 388-391] “The manipulation range of IHI (i.e., the difference between HIGH and LOW sessions) was 32.6 ± 30.7% (Cohen's *d* = 1.50) in all participants excluding 2 participants with data corruption (i.e., 22 participants).”

Line 263: "suggesting that the volitional control of SMR-ERDs in the closed-loop environment differs from spontaneous SMR fluctuations". This depends on how one defines the word "differs". In this case, the comparison is technically qualitative as the contrast is between a significant result (volitional control, p <0.05) and a non-significant one (spontaneous fluctuation, p > 0.05). However, a more direct comparison would involve comparing the two effects directly (volitional vs spontaneous) by statistically testing for a significant difference in effect sizes (e.g. using a two-way rmANOVA).Taking the effect size of the HIGH-vs-LOW contrast as an example, for volitional control this is d = 1.42, while for spontaneous fluctuation it is d = 0.49, which yields a differential d = 0.93 (1.42 -0.49). This is likely statistically significant but it will depend on the error term, hence the need for a formal two-way rmANOVA analysis with an interaction effect. This should be done for all the contrast analyses in the paper (including SMR power), instead of using the one-way ANOVA.

We appreciate your valuable comment. A direct statistical comparison was shown in Figure 4—figure supplements 2. Again, the "non-triggered TMS trials" were collections of the trials when SMR-ERD reached an equivalent threshold during the other two sessions. Therefore, we think that it is an understandable result that there is no significant interaction. We have revised the related parts in the manuscript as below.

Similarly, a two-way rmANOVA was applied when a direct statistical comparison is required (Figure 4—figure supplements 3C, D).

[Lines 455-554] “A two-way rmANOVA for sessions (three levels: HIGH, MID, and LOW) and trials (two levels: triggered TMS trial and non-triggered TMS trials) revealed a significant main effect for sessions (F_(2, 117)_ = 5.08, *p* = 0.008, η^2^ = 0.08), but no main effect for trials (F_(1, 117)_ = 0.32, *p* = 0.575, η^2^ < 0.01) and for interaction (F_(2, 117)_ = 1.82, *p* = 0.167, η^2^ = 0.03). A post-hoc two-tailed paired t-test showed significant differences between HIGH and LOW sessions (Cohen's *d* = 0.71, *p* = 0.006). Thus, no significant difference in IHI between the volitional control of SMR-ERDs in the closedloop environment differs from spontaneous SMR fluctuations was observed. A post-hoc two-tailed paired t-test showed significant differences between HIGH and MID sessions (Cohen's *d* = 0.94, *p* = 0.017), and between HIGH and LOW sessions (Cohen's *d* = 1.36, *p* < 0.001) in the triggered TMS trials. In contrast, in the non-triggered TMS trials, a post-hoc two-tailed paired t-test showed no significant differences between HIGH and MID sessions (Cohen's *d* = 0.20, *p* = 1.00), and between HIGH and LOW sessions (Cohen's *d* = 0.28, *p* = 1.00).”

[Lines 1850-1853] “To investigate the influence of spontaneous SMR fluctuations on IHI, a two-way rmANOVA for sessions (HIGH, MID, and LOW) and trials (triggered TMS trial and non-triggered TMS trial) as the withinparticipant factors.”

The dynamic range of SMR modulation occurs mainly in the ipsilateral hemisphere (in terms of bidirectional control). The contralateral hemisphere consistently shows increased SMR power relative to rest, with no bidirectional modulation. How do the authors explain this ? Could this be due to how the neurofeedback was implemented for each hemisphere (thresholds/rewards)? Right hand dominance? If the latter, where there any differential effects in the subgroup of left hand dominant participants?

We aimed to modulate the ipsilateral hemisphere bidirectionally while maintaining constant the contralateral excitability using our spatially bivariate BCI-based neurofeedback. To evaluate the effect of SMR modulation in the ipsilateral hemisphere, participants always aimed to maintain constant contralateral SMR-ERD. As written in the manuscript ("Spatially bivariate BCI-based neurofeedback"), visual feedback was provided on a computer screen in the form of cursor movements in a two-dimensional coordinate, in which x- and y-axis corresponded to the degree of the ipsilateral and contralateral SMR-ERD, respectively. For example, in HIGH session, participants were instructed to move the cursor toward the middle right (x > 0, y = 0) in the two-dimensional coordinate, the position showed a strong SMR-ERD in the ipsilateral hemisphere and moderate SMR-ERD in the contralateral hemisphere. On the contrary, in LOW session, participants were instructed to move the cursor toward the middle left (x < 0, y = 0) in the two-dimensional coordinate, the position showed a low SMR-ERD (or strong SMR-ERS) in the ipsilateral hemisphere and moderate SMR-ERD in the contralateral hemisphere. Therefore, spatially bivariate BCI-based neurofeedback enables us to modulate the ipsilateral SMR-ERD bidirectionally while maintaining constant contralateral SMR-ERD in the three sessions (i.e., HIGH, MID and LOW). In fact, Figures 2A and B showed the intended results. To help readers’ understanding, we prepared a sample video as Video 1.

The above is the answer to your point, but we notice that we need to describe the experimental limitation regarding (1) handedness and (2) the other patterns of the contralateral excitability. In this study, all participants were right-handed (Laterality Quotient: 72.2 ± 30.9%), as assessed by the Edinburgh Inventory (Oldfield, 1971), to eliminate bias due to the dominance. If a left-handed participant performs the same experiment, the modulation effect via BCI-based neurofeedback may be different. Moreover, in case the contralateral SMR-ERD was up- or downregulated instead of maintained at a moderate level, it is unclear whether the ipsilateral SMR-ERD was modulated bidirectionally and whether the IHI changed with it. The study performed a total of 11 sessions, including REST and NoFB, and the combinations of SMR-ERDs in both hemispheres (3 conditions vs. 3 conditions) may be worth evaluating in the future.

[Line 1325-1334] “For the limitations of the volitional control of SMR-ERD during unilateral MI, all participants were right-handed (Laterality Quotient: 72.2 ± 30.9%), as assessed by the Edinburgh Inventory (Oldfield, 1971), to eliminate bias due to the dominance. If a left-handed participant performs the same experiment, the modulation effect via BCI-based neurofeedback may be different. Moreover, in case the contralateral SMR-ERD was up- or downregulated instead of maintained at a moderate level, it is unclear whether the ipsilateral SMR-ERD was modulated bidirectionally and whether the IHI changed with it. The study performed a total of 11 sessions, including REST and NoFB, and the combinations of SMR-ERDs in both hemispheres (3 conditions vs. 3 conditions) may be worth evaluating in the future.”

Line 580: "the current technique can be tailored either to upregulate the damaged hemisphere, down-regulate the intact hemisphere…" and Line 585: "In conclusion, we presented an innovative approach to voluntarily and bidirectionally manipulate the state of IHI"Critically, bidirectional control would need to be significantly demonstrated above and below the resting baseline. Although the study shows bidirectional control of SMR power (in the ipsilateral hemisphere), this is strictly not the case for IHI (which shows only significant increases relative to rest). Hence, the term "bidirectional" should not be used in the context of the IHI, but rather the phrase "control of interhemispheric disinhibition". Hence, the authors could state that the current results should motivate future studies investigating whether upregulation of IHI in the damaged hemisphere is possible above and beyond spontaneous resting values.

We thank your important comments. Although we successfully demonstrated the up- and down-regulate the ipsilateral SMR-ERD (i.e., bidirectional changes) at EEG-level, the bidirectional changes in IHI from the right to the left hemisphere were not statistically shown since there is no significant difference in MID and LOW sessions (Figure 4B). Therefore, we have revised the expression of the “bidirectional” modulation in IHI and added the future experiment in the Limitation.

[Lines 1277-1282] “Although we successfully demonstrated the bidirectional changes in the ipsilateral SMR-ERD, the bidirectional changes in IHI from the right to the left hemisphere were not statistically significant since there was no significant difference between the MID and LOW sessions (Figure 4B). The current results should prompt future studies to ascertain whether down-regulation of IHI (i.e., disinhibition) during MI and up-regulation of IHI at rest in the non-dominant hemisphere (intact hemisphere) are possible in a similar setup.”

[Lines 1350-1355] “In conclusion, we presented an innovative approach to manipulate the state of IHI, by directly and bidirectionally modulating SMR-ERDs in a spatially bivariate BCI-based neurofeedback paradigm.”

Reviewer #3 (Recommendations for the authors):This is a well-conducted study building and expanding on the pioneering work of this group (Takemi et al. 2013, 2015). The novel contribution of this study is that self-regulated ERD is not modulating the TMS effects on the periphery (MEP) only, but also the TMS effects on the opposite brain hemisphere. This has been probed by IHI, but may probably have also been shown by TEP.All other interpretations with regard to "rebalancing", "pathway-specificity", "interhemispheric connectivity", and "challenging the concept of neurorehabilitation" etc. are far fetching and should be toned down. Such statements would have necessitated (i) either direct neurofeedback of interhemispheric coherence, (ii) or demonstration of a direct link between the self-regulated ERD and interhemispheric coherence, and (iii) the application in patient populations.Therefore, it is advisable to better link this study to the large body of literature in the IHI field, and discuss where this work adds additional insights. It is surprising, for example, that relevant previous literature is not referred to and discussed: Liang et al. (2014) have shown that imagined movements change IHI, Nelson et al. (2009) demonstrated bi-directional IHI, and Morishita et al. (2011, 2012) contributed with regard to ipsilateral inhibitory circuits and IHI. This work should be cited.

Thank you for your valuable comments and for introducing relevant previous studies. Although we agree that the parts pointed out need to be toned down, we would like to emphasize that we successfully revealed that modulating the excitability of the right hemisphere changes IHI from the right to the left hemisphere via BCI-based neurofeedback. This is fundamentally different from observational studies investigating movement-related IHI (Duque et al., 2007, 2005; Liang et al., 2014; Morishita et al., 2012; Murase et al., 2004; Nelson et al., 2009).

In addition, previous studies suggested that the IHI magnitude and the corticomotor excitability were modulated independently (Morishita et al., 2011, 2012). Although it is difficult to directly compare these results due to the significantly different motor tasks and experimental settings, the assertion that the projections from the transcallosal and the corticospinal pathways are distinct supports our findings (Figure 4—figure supplements 3B and C).

Furthermore, we discussed the neurophysiological mechanisms mediating IHI based on, for example, this literature (Carson, 2020). Regarding the results of interhemispheric coherence, we answered the next question.

We paid attention to the expression, and revised the manuscript as follows:

[Line 2] “Spatially bivariate EEG-neurofeedback can manipulate interhemispheric inhibition”

[Lines 1051-1056] “However, conventional BCI-based neurofeedback of the SMR signal from one hemisphere does not guarantee spatial specific activation of the sensorimotor activity in the targeted hemisphere (Buch et al., 2008; Caria et al., 2011; Soekadar et al., 2015a) because sensorimotor activities in the left and right hemispheres potentially influence one another, making IHI manipulation difficult.”

[Lines 45-47] “Our results show an interesting approach for neuromodulation, which might provide new treatment opportunities, for example, in patients suffering from a stroke. “

[Lines 1165-1168] “Conversely, under the conscious self-learning environment, volitional control of MEP amplitudes is retained for at least 6 months without further training (Ruddy et al., 2018), which supports the future prediction of long-term efficacy of IHI manipulation.”

[Lines 96-100] “A critical issue is whether a specific interhemispheric activity can be volitionally controlled. Previous studies have focused on inhibitory interhemispheric sensorimotor network, represented by interhemispheric inhibition (IHI), from movement-related manner (Duque et al., 2007, 2005; Liang et al., 2014; Murase et al., 2004; Nelson et al., 2009),”

[Lines 398-431] “In addition, we found disinhibition in NoFB session compared to REST session, although there was no statistical difference (Cohen's *d* = 0.42, *p* = 0.279); this trend is consistent with the previous studies, which reported that in healthy participants, IHI targeting the moving index finger leads to disinhibition before the movement onset to produce voluntary movement (Duque et al., 2005; Murase et al., 2004) and during isometric contraction (Nelson et al., 2009).”

[Lines 1307-1311] “Previous studies suggested that the IHI magnitude and the corticomotor excitability were modulated independently (Morishita et al., 2011, 2012). Although it is difficult to directly compare these results due to the significantly different motor tasks and experimental settings, the assertion that the projections from the transcallosal and the corticospinal pathways are distinct supports our findings.”

[Lines 1150-1155] “This may indicate that IHI and Network-intensity simply reflect the activity of different neural populations; interhemispheric projections act via surround/lateral inhibition in the sensory and motor cortices (Carson, 2020), while the synchrony at the EEG level reflects changes in the activity of local interactions between pyramidal neurons and interneurons in the thalamocortical loops (Pfurtscheller and Lopes da Silva, 1999).”

[Editors' note: further revisions were suggested prior to acceptance, as described below.]

Reviewer #1 (Recommendations for the authors):The authors provided very thorough and extensive revisions and they have addressed most of my comments. It is particularly reassuring to see the additional analyses of corticomotor excitability as probed by the CS.However, I'm still not completely convinced by the author's statistical approach.If I understood the approach correctly, % change in IHI and % change of CS MEP amplitude were entered into one rmANOVA model and the authors tested for a condition x index interaction effect (which did not reach significance). This is a rather strict comparison. A more traditional approach would be to add % CS change as a covariate of no interest and test whether the IHI effect survives even if the covariate is added. If that is the case, the authors have demonstrated that IHI changes over and above the variance explained by the CS effect which would support their initial hypothesis that IHI can be modulated via their EEG-BCI and is not simply an epiphenomenon. No other statistical analyses would be required. However, the discussion would need to be modified!The authors also analysed changes in CS MEP amplitude across conditions. CS MEPs did not significantly differ across the different EEG-BCI conditions. However, the absence of a significant effect does not allow the authors to conclude that there is "no effect". This would require a Bayesian approach which would test whether the data provides evidence in favour of H0. This could be another argument that IHI changes over and above CS changes. However, I don't think that these additional analyses would be needed if the co-variance analysis of IHI would reach significance. Maybe the authors might want to consult a stats expert for this analysis.

We appreciate the suggestion to use an alternative statistical approach. According to the suggestion, we tested whether the effect of IHI manipulation remains significant after a covariate is added. As a result, we confirmed that there was a significant difference in IHI magnitude between sessions (*p* < 0.001) using a generalized linear model with the percent changes of left MEP amplitude as a covariate of no interest (Figure 4—figure supplements 3C). Therefore, we did not analyze the left MEP amplitude, and only displayed the results (Figure 4—figure supplements 3B). Based on this, we successfully demonstrated that IHI changes greater than the variance explained by the CS effect and manipulation is not simply an epiphenomenon. In addition, we analyzed the non-triggered trials to investigate the CS effect (Figure 4—figure supplements 3D). During the non-trigged trials, no significant difference was observed in IHI magnitude (*p* = 0.319).

Based on the findings that IHI can be manipulated via BCI-based neurofeedback, further experiments may help to better understand the pathway specificity. In future studies, to further confirm whether the participants learned pathway-specific IHI manipulation, the CS effect should be kept constant across the conditions by adjusting the CS intensity to evoke left MEP of 1 mV peak-to-peak amplitude.

We have revised the manuscript to include these additional analyses, figures, and discussion.

[Lines 337-354] “Although we revealed that modulating the excitability of the right hemisphere changes IHI magnitude from the right to the left hemisphere, the specificity of the IHI manipulation was not proved yet. We presented additional results related to changes in corticospinal excitability as assessed by MEPs in the left FDI in response to the CS over the right hemisphere (Figure 4—figure supplements 3B).

To investigate whether changes in IHI from the right to the left hemisphere and changes in corticospinal excitability in the right hemisphere are independent phenomena, a generalized linear model with the percent changes of left MEP amplitude as a covariate of no interest was applied. This approach tests whether the effect of IHI manipulation survives even if the covariate is added. During the triggered TMS trials, a generalized linear model for sessions (three levels: HIGH, MID, and LOW) revealed a significant main effect for sessions (F_(3, 60)_ = 12.75, *p* < 0.001, η^2^ = 0.44). Post-hoc two-tailed paired t-test in the increase in IHI showed significant differences between HIGH and MID sessions (Cohen's *d* = 1.06, *p* < 0.001), and between HIGH and LOW sessions (Cohen's *d* = 1.42, *p* < 0.001). During the non-triggered TMS trials, a generalized linear model for sessions (three levels: HIGH, MID, and LOW) revealed no main effect for sessions (F_(3, 60)_ = 1.13, *p* = 0.319, η^2^ = 0.10). Therefore, we successfully demonstrated that IHI changes greater than the variance explained by the CS effect and manipulation is not simply an epiphenomenon.”

[Lines 647-652] “In addition, we successfully demonstrated that IHI changes greater than the variance explained by the CS effect. Based on our findings, further experiments may help to better understand the pathway specificity. To further confirm whether the participants learned pathway-specific IHI manipulation, the CS effect should be kept constant across the conditions by adjusting the CS intensity to evoke left MEP of 1 mV peak-to-peak amplitude.”

[Lines 1122-1125] “To investigate whether changes in IHI and corticospinal excitability in response to the CS over the right hemisphere are independent phenomena, a generalized linear model with the percent changes of left MEP amplitude as a covariate of no interest was conducted for the triggered and non-triggered TMS trials.”

Reviewer #3 (Recommendations for the authors):I am satisfied with the author's responses and revisions, which are extensive and much more consistent with the collected evidence from the experiment.Ultimately, I am ambivalent (i.e. neither for nor against) on recommending this paper for publication in eLife based on its scientific significance. On a positive note, the authors clearly demonstrate that it is possible to modulate IHI (albeit unidirectionally) using bivariate neurofeedback, and this is an original and scientifically novel contribution. On the other hand, from the experiments conducted, it is not clear whether the bivariate neurofeedback approach is critical for the observed findings since there is no comparison to the conventional BCI-based neurofeedback that exploits SMR-ERD from one hemisphere. Hence the results show evidence of "sufficient" conditions to reduce inter-hemispheric inhibition (IHI) but do not provide evidence of "necessary" conditions i.e. whether uni-hemispheric neurofeedback would lead to similar effects but with a much more parsimonious experimental setup that does not involve controlling EEG from both hemispheres.I, therefore, leave it to the senior editor for the ultimate decision regarding publication, taking also into account the other reviewers' comments.

We have added an in-depth comparison of the new bivariate and univariate neurofeedback methods.

First, this study did not perform to show the necessary and sufficient condition that IHI can be manipulated only via bivariate BCI-based neurofeedback. However, we emphasize that our results are still significant because this was the first study to show IHI-state manipulation induced by volitional variation in ipsilateral sensorimotor excitability to the imagined hand, which has not been achieved with the conventional univariate BCI-based neurofeedback paradigm.

A direct comparison between bivariate and univariate (i.e., ipsilateral hemisphere to the imagined hand) BCI-based neurofeedback might help solidifying the necessity of bivariate BCI-based neurofeedback for targeted IHI manipulation, but it was not feasible to incorporate experiments in the present study due to experimental time limitation (the experimental time would need to be doubled to 5–6 h to perform such an experiment, violating ethical concerns). A future study should be performed to directly compare the effect size of the IHI manipulation between bivariate and univariate BCI-based neurofeedback in a within-participant cross-over design with adjusting experimental time by omitting some experimental conditions. Again, the present study focused to show the feasibility of bivariate BCI-based neurofeedback for targeted IHI manipulation, and required 2.5–3 h of the experimental protocol for proving this.

When conducted a future experiment for direct comparison of bivariate and univariate BCI-based neurofeedback, it is highly probable that the necessary conditions will be satisfied for the following reasons.

Our findings indicated that unilateral (not bilateral) up- or down-regulation of the sensorimotor excitability manipulates the IHI magnitude (Figure 2C and Figure 4B), which is consistent with previous studies (Carson, 2020; Duque et al., 2005; Murase et al., 2004). However, conventional methods did not guarantee unilateral specific sensorimotor activation (Buch et al., 2008; Caria et al., 2011; Soekadar et al., 2015). In addition, SMR-ERD was observed focused over the scalp sensorimotor areas, mainly bilaterally (e.g., Pichiorri et al., 2011; Vidaurre et al., 2019). Therefore, the effect of IHI modulation may be minimal because the bilateral up-regulation of the sensorimotor excitability via univariate BCI-based neurofeedback may inhibit the other hemispheres. We did not investigate whether the conventional univariate neurofeedback that exploits SMR-ERD from one hemisphere achieves bidirectional modulation of the target-hemisphere-dependent SMR-ERD and the accompanying IHI magnitude, but it is assumed that the sensorimotor excitability in the opposite hemisphere would be also up-regulated through univariate neurofeedback, leading to a weaker range of IHI manipulation.

Based on the aforementioned factors, we presumed that it is difficult to produce a similar effect as this study via univariate BCI-based neurofeedback. Even if IHI-state manipulation is equally achieved via univariate BCI-based neurofeedback, our scientific significance remains established. Direct comparison would be executed beyond this work.

According to the reviewer’s suggestion, we have revised the discussion and limitations in the manuscript as follows:

[Lines 709-723] “In addition, it is not clear from previous experiments whether the bivariate neurofeedback approach is superior to the conventional BCI-based neurofeedback that exploits SMR-ERD from one hemisphere. Since both hemispheres are connected by intrinsic transcallosal projections and exhibit functional crosstalk (Arai et al., 2011; Hofer and Frahm, 2006; Meyer et al., 1995; Waters et al., 2017), it is expected that the sensorimotor excitability in the opposite hemisphere would also be up-regulated through univariate (i.e., ipsilateral hemisphere to the imagined hand) BCI-based neurofeedback. Consistent with the findings that the unilateral up- or down-regulation of the sensorimotor excitability manipulates the IHI magnitude, such sensorimotor activation without spatial specificity may inhibit the other hemisphere, leading to weaker range of IHI manipulation. Thus, we presumed that it is difficult to produce a similar effect as this study via univariate BCI-based neurofeedback. A future study is required to directly compare the effect size of IHI manipulation between bivariate and univariate BCI-based neurofeedback in a within-participant cross-over design with adjusting experimental time by omitting certain experimental conditions.”

**References**

Arai N, Müller-Dahlhaus F, Murakami T, Bliem B, Lu M-K, Ugawa Y, Ziemann U. State-dependent and timingdependent bidirectional associative plasticity in the human SMA-M1 network. *J Neurosci* 31(43), 15376–15383, 2011.

Buch E, Weber C, Cohen LG, Braun C, Dimyan MA, Ard T, Mellinger J, Caria A, Soekadar S, Fourkas A, Birbaumer N. Think to move: A neuromagnetic brain-computer interface (BCI) system for chronic stroke. *Stroke* 39(3), 910–917, 2008.

Caria A, Weber C, Brötz D, Ramos A, Ticini LF, Gharabaghi A, Braun C, Birbaumer N. Chronic stroke recovery after combined BCI training and physiotherapy: A case report. *Psychophysiology* 48(4), 578–582, 2011.

Carson RG. Inter-hemispheric inhibition sculpts the output of neural circuits by co-opting the two cerebral hemispheres. *The Journal of Physiology* 598(21), 4781–4802, 2020.

Duque J, Hummel F, Celnik P, Murase N, Mazzocchio R, Cohen LG. Transcallosal inhibition in chronic subcortical stroke. *Neuroimage* 28(4), 940–946, 2005.

Hofer S, Frahm J. Topography of the human corpus callosum revisited: Comprehensive fiber tractography using diffusion tensor magnetic resonance imaging. *NeuroImage* 32(3), 989–994, 2006.

Meyer B-U, Röricht S, von Einsiedel HG, Kruggel F, Weindl A. Inhibitory and excitatory interhemispheric transfers between motor cortical areas in normal humans and patients with abnormalities of the corpus callosum. *Brain* 118(2), 429–440, 1995.

Murase N, Duque J, Mazzocchio R, Cohen LG. Influence of interhemispheric interactions on motor function in chronic stroke. *Ann Neurol* 55(3), 400–409, 2004.

Pichiorri F, Fallani FDV, Cincotti F, Babiloni F, Molinari M, Kleih SC, Neuper C, Kübler A, Mattia D. Sensorimotor rhythm-based brain–computer interface training: The impact on motor cortical responsiveness. *J Neural Eng* 8(025020), 9, 2011.

Soekadar SR, Birbaumer N, Slutzky MW, Cohen LG. Brain–machine interfaces in neurorehabilitation of stroke.

*Neurobiology of Disease* 83(1), 172–179, 2015.

Vidaurre C, Ramos Murguialday A, Haufe S, Gómez M, Müller K-R, Nikulin VV. Enhancing sensorimotor BCI performance with assistive afferent activity: An online evaluation. *NeuroImage* 199375–386, 2019.

Waters S, Wiestler T, Diedrichsen J. Cooperation not competition: Bihemispheric tDCS and fMRI show role for ipsilateral hemisphere in motor learning. *J Neurosci* 37(31), 7500–7512, 2017.